# Cytosolic peptides encoding Ca$_V$1 C-termini downregulate the calcium channel activity-neuritogenesis coupling

Yaxiong Yang [1,2,8], Zhen Yu[1,2,8], Jinli Geng[1,2,8], Min Liu[3], Nan Liu[4], Ping Li[1], Weili Hong [1], Shuhua Yue [1], He Jiang[3], Haiyan Ge[3], Feng Qian[5], Wei Xiong[6], Ping Wang[7], Sen Song[3], Xiaomei Li [3✉], Yubo Fan [1✉] & Xiaodong Liu [1,2✉]

L-type Ca$^{2+}$ (Ca$_V$1) channels transduce channel activities into nuclear signals critical to neuritogenesis. Also, standalone peptides encoded by Ca$_V$1 DCT (distal carboxyl-terminus) act as nuclear transcription factors reportedly promoting neuritogenesis. Here, by focusing on exemplary Ca$_V$1.3 and cortical neurons under basal conditions, we discover that cytosolic DCT peptides downregulate neurite outgrowth by the interactions with Ca$_V$1's apo-calmodulin binding motif. Distinct from nuclear DCT, various cytosolic peptides exert a gradient of inhibitory effects on Ca$^{2+}$ influx via Ca$_V$1 channels and neurite extension and arborization, and also the intermediate events including CREB activation and c-Fos expression. The inhibition efficacies of DCT are quantitatively correlated with its binding affinities. Meanwhile, cytosolic inhibition tends to facilitate neuritogenesis indirectly by favoring Ca$^{2+}$-sensitive nuclear retention of DCT. In summary, DCT peptides as a class of Ca$_V$1 inhibitors specifically regulate the channel activity-neuritogenesis coupling in a variant-, affinity-, and localization-dependent manner.

[1] Key Laboratory for Biomechanics and Mechanobiology of Ministry of Education, Beijing Advanced Innovation Center for Biomedical Engineering, School of Biological Science and Medical Engineering, School of Engineering Medicine, Beihang University, Beijing 100083, China. [2] X-Laboratory for Ion-Channel Engineering, Beihang University, Beijing 100083, China. [3] School of Medicine, Tsinghua University, Beijing 100084, China. [4] Center for Life Sciences, School of Life Sciences, Yunnan University, Kunming 650091, China. [5] School of Pharmaceutical Sciences, Tsinghua University, Beijing 100084, China. [6] School of Life Sciences, Tsinghua University, Beijing 100084, China. [7] Laboratory for Biomedical Engineering of Ministry of Education, Zhejiang University, Hangzhou 310027, China. [8] These authors contributed equally: Yaxiong Yang, Zhen Yu, Jinli Geng. ✉email: li-xiaomei@mail.tsinghua.edu.cn; yubofan@buaa.edu.cn; liu-lab@buaa.edu.cn

Voltage-gated $Ca^{2+}$ channels ($Ca_V$) are closely involved in diverse pathophysiological processes, by generating $Ca^{2+}$ signals in response to membrane potentials[1,2]. L-type $Ca^{2+}$ channels ($Ca_V1$) from the $Ca_V$ family, $Ca_V1.2$ and $Ca_V1.3$ in particular, are widely expressed in human tissues and organs including the nervous system[3]. Among its multiple roles, $Ca_V1$ channels mediate the signaling cascade known as excitation-transcription coupling, which transduces cellular stimuli into nuclear signals to regulate transcription of essential genes manifested into the growth conditions of neurites, which constitutes $Ca_V1$-dependent excitation-neuritogenesis coupling[4–7]. Notably, such coupling between $Ca_V1$ channels and neuritogenesis signaling is also functional even under basal conditions since $Ca^{2+}$ channels are still active, e.g., to mediate the slow calcium oscillations[8,9]. That says, the essential factor to couple the downstreams is the activities of $Ca_V1$ channels, either at low (basal) or high (excited) levels, which explains the up- or downregulations of $Ca_V1$-mediated signaling and neuritogenesis upon high $K^+$ stimulation and other perturbations[10,11]. Both $Ca_V1$ functions and neurite development are closely involved in learning and memory, and a broad spectrum of mental disorders[12,13]. The cascade linking $Ca_V1$ to neuritogenesis is coordinated by a number of key $Ca^{2+}$-signaling proteins including calmodulin (CaM), $Ca^{2+}$/CaM-dependent Kinase II (CaMKII), and calcineurin, to accomplish: $Ca_V1$ channel gating and $Ca^{2+}$ influx, cytonuclear translocation of key molecules CaM and CaMKII, activation of nuclear transcription factors such as CREB (cAMP-response element-binding protein) and NFAT (nuclear factor of activated T-cells), expression of critical genes including c-Fos, and ultimately branching and elongation of neurites[10,14–18]. In parallel, the peptide fragments encoded by distal carboxyl-terminus (DCT) of $Ca_V1$ could also act as transcription factors in the nucleus that directly regulate gene transcription and expression, to promote neurite outgrowth[19].

The effects of DCT as the key domain of $Ca_V1$ have been well characterized. By competing with apoCaM ($Ca^{2+}$-free calmodulin), intramolecular DCT autonomously binds the canonical CaM-binding motif (the preIQ-IQ domain of $Ca_V1$), causing concurrent effects on channel gating, i.e., weaker $Ca^{2+}$-dependent inactivation and reduced voltage-gated activation[20–23]. Chemical-induced dimerization of the proximal- and distal-DCT subdomains demonstrated that the reduction of $Ca^{2+}$ influx is solely due to acute binding of DCT, ruling out other potential mechanisms of action[21,24,25]. Effects of DCT on $Ca_V1$ are in direct opposition to apoCaM, consistent with a mechanism of strict competition between DCT and apoCaM[20,23]. As summarized by the term CMI (C-terminus mediated inhibition)[24], DCT is able to produce multifaceted and coherent effects, including competitive binding against apoCaM, reduction of $Ca^{2+}$ influx, and concurrent attenuation of inactivation and activation.

Besides intramolecular DCT inhibition, standalone DCT peptides have also been reported to bind and inhibit $Ca_V1$ channels[20–22,26–31]. We postulated that DCT peptides through a mechanism of intermolecular CMI should attenuate $Ca^{2+}$/$Ca_V1$ signaling to the nucleus. In support, some $Ca_V1$ inhibitors such as dihydropyridine (DHP) do exert inhibitory effects on $Ca_V1$-dependent signaling and neuritogenesis[10,11,32]. We pursued the above hypothetical inhibition on $Ca_V1$-dependent neuritogenesis by standalone DCT peptides, which, however, would be in direct contradiction to neuritogenic DCT effects as demonstrated by CCAT (Calcium Channel Associated Transcription regulator, encoded by $Ca_V1.2$ DCT)[19], thus encountered with an immediate dilemma. In this work, we are motivated by such peptide CMI that supposedly downregulates the $Ca_V1$-neuritogenesis coupling through a unique mechanism of action: DCT peptides bind $Ca_V1$ at its CaM-binding motif.

DCT peptides have been found endogenously in native cells under physiological conditions[33]. These peptides could result from distinct production mechanisms in various cell types, and may have different lengths or compositions, but are all encoded by DCT fragments of high homology across the $Ca_V1$ family (Supplementary Fig. 1 and Supplementary Table 1). In skeletal muscle cells, $Ca_V1.1$-encoded peptides of ~30-40 kDa are produced by proteolysis[34–36], named as $CCT_S$ (Cleaved Carboxyl-Terminal fragment from $Ca_V1.1$ pore-forming subunit $\alpha_{1S}$; and $CCT_C$ and $CCT_D$ from $\alpha_{1C}$ and $\alpha_{1D}$, respectively). Similarly, $CCT_C$ is cleaved from $Ca_V1.2$ in neurons[37] or cardiac myocytes[38,39]. Presumably by $Ca_V1.3$ cleavage, $CCT_D$ of ~40 kDa was found in cardiac myocytes[40]. Besides proteolytic cleavage, DCT peptides could also be generated by bicistronic mechanisms via exonic promoters for direct translation[19,41–43], e.g., $CCAT_C$ of ~60 kDa or more. In summary, two types of DCT peptides (~40 kDa and ~60 kDa) have been evidenced from diverse preparations[33], alongside with a shorter peptide of ~15 kDa in neurons (encoded by the last ~100 a.a. at the C-terminal end of $Ca_V1.2$)[41]. Substantial discrepancies exist among various peptides in sequences, mechanisms of production, specificity to cell types, and effects on channels or cells. In this work, we undertook the task to clarify the actual roles of DCT-encoded peptides, focusing on the hypothesis that these peptides of diverse forms essentially share the same principle: affinity-dependent binding of $Ca_V1$ channels to downregulate $Ca_V1$ activities and channel activity-dependent neuritogenesis.

## Results

**Hints on DCT peptide inhibition of $Ca_V1$ channels and neuritogenesis.** We first conducted sequence alignment for homologous DCT domains across $Ca_V1.1$-$1.4$, containing the major fragments of proximal C-terminal regulatory domain (PCRD), nuclear retention domain (NRD) and distal C-terminal regulatory domain (DCRD) (Supplementary Fig. 1 and Supplementary Table 1). The NRD motif is an indispensable region for $Ca^{2+}$-dependent nuclear export as demonstrated in $CCAT_C$[19] and $CCT_D$[40]. The PCRD and DCRD cooperate to compete with apoCaM for binding the IQ motif of $Ca_V1$ as the molecular basis of CMI, where the DCRD plays a dominant role compared with the PCRD[22,24]. Two transcription activation domains are localized in the PCRD-NRD junction and the DCRD motif[19,41], respectively. Based on these and earlier analyses, the representative peptides of three major categories have been focused on, including 1) DCT peptides (~60 kDa) via bicistronic transcription, such as $CCAT_C$ that contains the entire DCT; 2) $CCT_C$ or $CCT_D$ (~40 kDa) from posttranslational cleavage lacking the PCRD domain but still incorporating the majority of DCT (from NRD to DCRD); 3) the short peptide of DCRD (~15 kDa) that is sufficient to modulate $Ca_V1$ gating, although its physiological relevance is relatively less established.

As proof of principle, the $DCRD_F$ was overexpressed in cortical neurons, considering that this short peptide encoded by the last ~100 a.a. of $Ca_V1.4$ DCT has been thoroughly characterized for its strong competition with apoCaM to bind onto the channel[20,24,27,31]. By the cocktail treatments of cultured cortical neurons, the relative contribution of $Ca_V1.3$ channels to $Ca_V1$ currents was evaluated (see Methods and Supplementary Fig. 2 for details). Based on the patch-clamp recordings by a voltage ramp and the representative step of $-10$ mV, $Ca_V1.3$ made a significant contribution to the total $Ca_V1$ currents in cortical neurons (~50%), in agreement with the previous reports suggesting that both $Ca_V1.3$ and $Ca_V1.2$ are critical to $Ca_V1$ signaling in cortical neurons and hippocampal neurons[44–48]. With the full cocktail recipe, $Ca^{2+}$ currents

mainly mediate by $Ca_V1.3$ were isolated and recorded to examine the effects of $DCRD_F$ peptides (Fig. 1a). The $DCRD_F$ potently attenuated cortical $Ca_V1.3$ currents at the peak; meanwhile, the steady-state amplitude (measured at 300 ms) was nearly unchanged. Such characteristic effects on native $Ca_V1.3$ channels are highly consistent with the CMI modulation of recombinant $Ca_V1.3$ channels[24], where reduction of $Ca^{2+}$ influx is ensured by concurrent attenuation of activation and inactivation evidenced from both acute and long-term effects. We then examined the hypothetical role of DCT inhibition on $Ca_V1$-dependent neuritogenesis signaling. As expected, the $DCRD_F$ peptide caused a significant reduction in neurite outgrowth and branching of cortical neurons under basal conditions, as measured by the total length and complexity (Sholl analysis) respectively (Fig. 1b). In contrast, the mutant peptide $DCRD_F\_V/A$ (V/A denotes V41A, a loss-of-function

mutation) produced no damage on neurite outgrowth of cortical neurons overexpressing $DCRD_F$.

We further checked the major signals along the well-established cascade, including the phosphorylation of a key transcription factor CREB[49,50]. $Ca_V1$ would be the major path of $Ca^{2+}$ entry preferred by downstream CaMKII/CREB signaling over $Ca_V2$[11], $DCRD_F$ peptides strongly attenuated pCREB signals (immunostaining of phosphorylated CREB or pCREB, Fig. 1c). In contrast, pCREB exhibited no difference between the control neurons and the mutant group $DCRD_F\_V/A$[20,24]. Furthermore, the expression level of c-Fos, one of the classical immediate early genes driven by pCREB[15,16], was significantly reduced by $DCRD_F$ but not $DCRD_F\_V/A$ (Fig. 1d). Additional stimulation to enhance membrane excitation or channel activation is expected to provide higher dynamic ranges, although the $Ca_V1$ channel activity-neuritogenesis coupling should function similarly in cortical

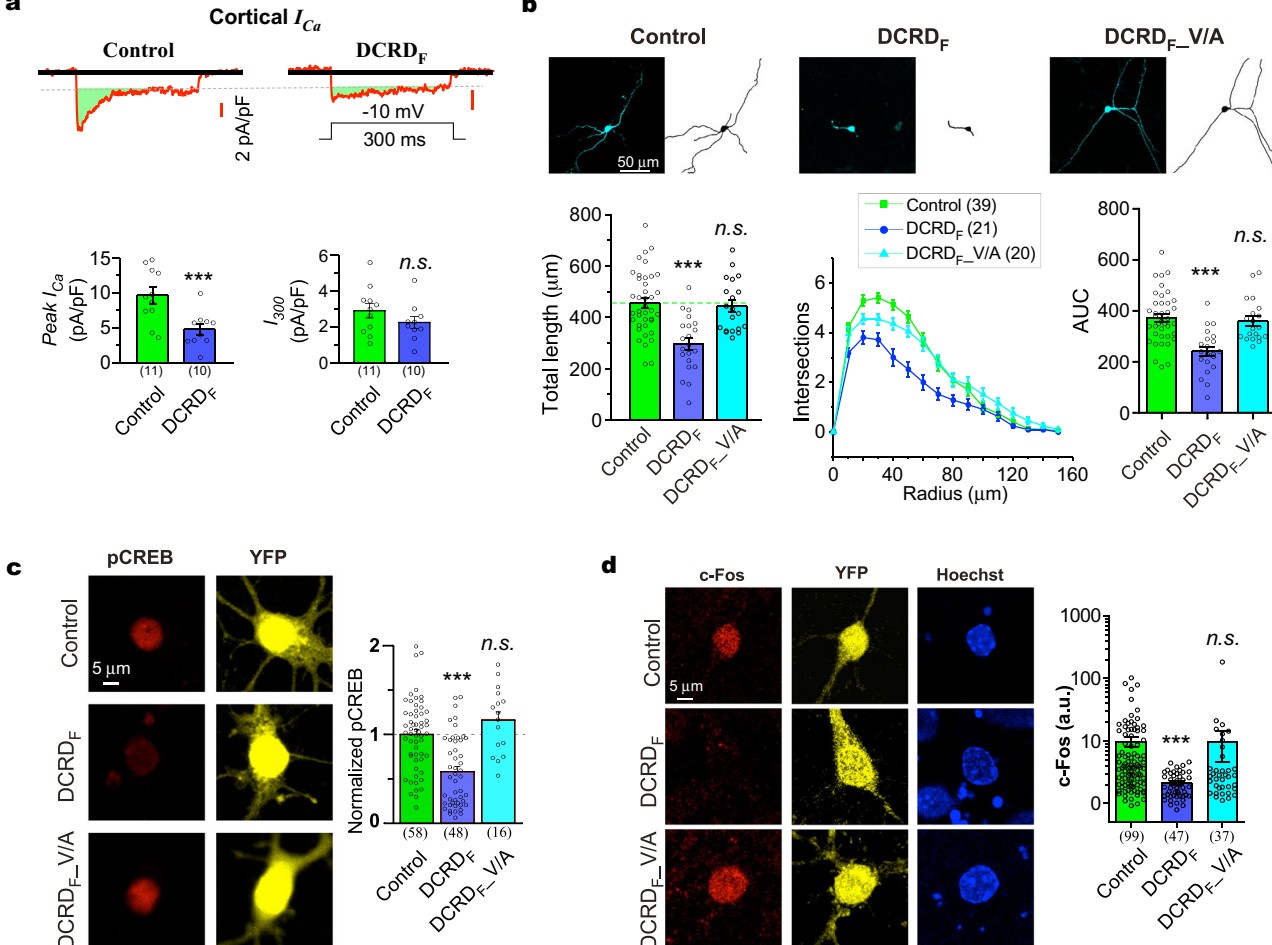

**Fig. 1 $DCRD_F$ effects on $Ca_V1.3$ gating and neuritogenesis signaling in cortical neurons. a** Inhibition of endogenous $Ca_V1.3$ currents by $DCRD_F$ peptides. Cultured neurons were treated with the cocktail recipe to isolate $Ca_V1.3$ $Ca^{2+}$ current (details in Supplementary Fig. 2). Exemplary $Ca^{2+}$ current (upper, scale bar in red) was elicited by the standard protocol of voltage step (300 ms; -10 mV). Potential inhibitory effects on channel functions were evaluated by the indices of peak and steady-state current amplitudes (pA/pF, bottom). **b** Effects on neuronal morphology. Based on the confocal fluorescent images of peptide-expressing cortical neurons (upper row), neurite tracing for each neuron was performed (middle row), total neurite length and Sholl analyses summary (bottom row) were compared among the three groups: YFP, YFP-$DCRD_F$ and YFP-$DCRD_F\_V/A$. Sholl analyses are routinely accompanied by the quantitative index of AUC (area under the curve, lower right). **c** Effects of $DCRD_F$ peptides on pCREB signals. Cortical neurons were transfected with YFP, YFP-$DCRD_F$ or YFP-$DCRD_F\_V/A$ (loss-of-function mutant), respectively. The pCREB signals were evaluated by immunofluorescence. Red and yellow fluorescence in confocal images represent pCREB signals and overexpressed YFP, respectively. pCREB signals were normalized over the YFP control group. **d** Effects on c-Fos signals. Cortical neurons expressing YFP, YFP-$DCRD_F$ or YFP-$DCRD_F\_V/A$ were stained with c-Fos antibody (in red). Fluorescent intensities of c-Fos signals in the nuclei were summarized. Student's *t*-test (**a**), one-way ANOVA followed by Bonferroni for post hoc tests (**b, c**) and Kruskal-Wallis and Dunn's non-parametric test (**d**, non-normal distribution, checked by D'Agostino & Pearson omnibus normality test) were used (***$p < 0.001$; *n.s.*, not significant, $p > 0.05$). Values are represented as mean ± SEM.

neurons of either conditions (basal or excited). Indeed, when the neurons were stimulated by 40 mM K$^+$, similar results from WT and mutant DCRD$_F$ were obtained when we reexamined the peptide effects on pCREB signals (Supplementary Fig. 3a). In addition, the enhanced signals by high K$^+$ stimulation provided the opportunity to capture the potent inhibition of DCRD$_F$ on the translocation of CaM (from Cytosol to Nucleus, defined by N/C ratio, Supplementary Fig. 3b), another key event along the Ca$_V$1-triggered signaling pathway[49,50]. Here, the observed effects arise from DCRD$_F$ inhibition on Ca$_V$1 channels which can be well represented by Ca$_V$1.3, especially in cortical neurons under basal conditions.

**CMI effects on recombinant Ca$_V$1.3 channels by various DCT peptides.** Encouraged by the results that DCT peptides downregulated cortical Ca$_V$1 channel-dependent transcription and neuritogenesis, we proceeded further with the recombinant Ca$_V$1.3 channels for the details on DCT effects. We chose five variants encoded by DCT of Ca$_V$1.2 or Ca$_V$1.3 which represent the native forms of DCT peptides: one long-form variant CCAT$_C$ (~60 kDa), two medium-form variants CCT$_C$ and CCT$_D$ (~40 kDa), and two short-form variants DCRD$_C$ and DCRD$_D$ (~15 kDa) (Fig. 2a). To quantify their effects on the gating of full-length Ca$_V$1.3 ($\alpha_{1DL}$) channels, the two major indices were routinely examined: inactivation (the strength of Ca$^{2+}$-dependent inactivation, $S_{Ca}$) and activation (the peak amplitude of Ca$^{2+}$ current, $I_{Ca}$)[24] (Fig. 2b, left column). Firstly, consistent with the previous report[27], the DCRD$_F$ peptides generated characteristic CMI effects: concurrent attenuation of both inactivation and activation, as illustrated by the altered profiles (the green shades to illustrate the actual attenuation) (Supplementary Fig. 4a). In direct contrast, DCRD$_{F}$_V/A did not cause any appreciable change in gating indices ($S_{Ca}$ or $I_{Ca}$). CMI effects on activation were also evidenced from Ba$^{2+}$ currents which were significantly inhibited by DCRD$_F$ but not by DCRD$_{F}$_V/A (Supplementary Fig. 4b). Notably, demonstrated by voltage-dependent steady-state (at 300 ms) currents, DCRD$_F$ and DCRD$_{F}$_V/A peptides are essentially indistinguishable from the $\alpha_{1DL}$ control, supporting the notion that intermolecular CMI (by standalone DCT peptides) shares similar mechanisms with intramolecular CMI (by the DCT motif covalently-linked to the channel), supposedly in an acute manner as previously proved[24].

Following the initial evidence from the Ca$_V$1.4 DCRD$_F$, we performed a systematic comparison for the five representative peptides (Fig. 2b), each of which was co-expressed with Ca$_V$1.3 channels. To better quantify DCT effects, the inhibition potency (*CMI*, in percentage) is defined as the normalized fraction of channels that switch from apoCaM-bound to DCT-bound, which can be directly calculated from the inactivation parameter $S_{Ca}$ before and after peptide inhibition (Eqs. E2 and E3 in Methods). *CMI* is inversely proportional to $S_{Ca}$, therefore strong DCT such as DCRD$_F$ should have higher *CMI* than weaker DCT such as DCRD$_{F}$_V/A.

By YFP fluorescence intensities, individual cells expressing YFP-tagged DCT were scrutinized to ensure that the expression levels of these peptides were at the comparable (high) levels. DCT peptides of DCRD$_C$, CCT$_C$, DCRD$_D$ and CCT$_D$ clearly produced CMI effects of substantial potency, except that CCAT$_C$ only slightly attenuated Ca$_V$1.3 gating (weak *CMI*) (Fig. 2c). Electrophysiological profiling of peptide CMI lays the foundation for our subsequent investigations into the effects of DCT peptides on Ca$_V$1-dependent neuritogenesis. CCAT$_C$ exhibited rather weak (insignificant) effects on channel gating, which appears to agree with the earlier report where the long DCT$_C$ was found to have no effect on Ca$^{2+}$-dependent inactivation of Ca$_V$1.2[22]. However,

CCT$_C$, the shorter motif encoded by a portion of CCAT$_C$, is capable of strong CMI. Also, the key segment DCRD$_C$ has the capability to attenuate Ca$^{2+}$-dependent inactivation. In this work, one of our aims is to clarify the above discrepancy regarding various DCT$_C$ peptides. In fact, all the four DCT domains across the Ca$_V$1 family are homologous (including the DCRD domains), suggesting high similarities in their functional roles (Supplementary Fig. 1). Taking the DCRD$_F$ as the exemplar, its core segment was further narrowed down to DCDR$_{F}$_17-66 (the residues between S17 and L66) well conserved among Ca$_V$1.2-1.4 (but not Ca$_V$1.1), which may account for the potent inhibition observed from DCRD$_C$, DCRD$_D$ or DCRD$_F$ peptides (Supplementary Fig. 5).

**Ca$_V$1 channels and neuritogenesis are inhibited by cytosolic DCT peptides.** In the context of Ca$_V$1-dependent neuritogenesis, modulation of Ca$_V$1 channels would make changes in the growth of neurites, provided that the channel activity-transcription coupling is coherently regulated. Based on the potent inhibition by DCRD$_F$ (Fig. 1), we pursued the hypothesis further that the DCT peptides of native forms may induce inhibitory effects in accordance with CMI potency (Fig. 2c). Since the signals or events at all the checkpoints (Ca$_V$1 gating, Ca$^{2+}$ influx, CaM translocation, pCREB, c-Fos and neuritogenesis) were consistently attenuated by DCRD$_F$ in cortical neurons (Fig. 1), Ca$_V$1 gating and neurite growth, the major input and output respectively, were selected as the two major checkpoints to represent the full cascade of channel activity-neuritogenesis coupling. By overexpressing DCT peptides in cortical neurons with careful scrutinization of cellular fluorescence as in electrophysiology (Fig. 2), the potential CMI on neuritogenesis was examined for the representative peptides of CCAT$_C$, CCT$_C$, and CCT$_D$ (long and medium forms), along with the peptides DCRD$_C$ and DCRD$_D$ (short form). To our surprise, statistically none of CCAT$_C$, CCT$_C$, and CCT$_D$ exhibited any significant effect on neurite length and branching even for ensured overexpression (Fig. 3a–c). Meanwhile, resembling DCRD$_F$ inhibition, DCRD$_C$ and DCRD$_D$ peptides induced significant neurite retractions. Since functional Ca$_V$1 channels are located at the plasma membrane, CMI effects by DCRD peptides should take place in the cytosol. We then examined the cytosolic-nuclear distribution for each peptide variant, indexed by its N/C ratio (Fig. 3d). On average, the N/C ratio values for DCRD$_C$ and DCRD$_D$ fell below the control level (YFP, with N/C ratio ~1.5)[51], showing a pattern of cytosolic distribution; in contrast, the peptides CCAT$_C$, CCT$_C$, and CCT$_D$ were more distributed into the nucleus (N/C ratio>1.5). For each variant, by applying N/C ratio criteria (cut-off value of 1.5) the neurons could be divided into two distinct (cytosolic versus nuclear) subgroups. For the cytosolic subgroup, similar to DCRD$_C$ and DCRD$_D$, cytosolic CCT$_C$ and CCT$_D$ significantly attenuated neurite outgrowth (Fig. 3e, f). Notably, although cytosolic CCAT$_C$ appeared to have a tendency of attenuation, its actual effects on neurites turned out to be rather mild with no statistical significance, consistent with its weak CMI on Ca$_V$1.3 gating. We then postulated that cytosolic DCT peptides would downregulate the Ca$_V$1-dependent neuritogenesis. In support, for the five representative peptides we tested in cortical neurons, an inverse correlation appeared to exist between CMI potency of cytosolic peptides and neurite length (Fig. 3g). Hence, it is likely that the DCT peptides present in the cytosol share the same mechanisms with DCRD$_F$ to inhibit Ca$_V$1 gating and signaling (Fig. 1).

**Both PCRD and DCRD tune CMI potency of DCT variants.** No structural information of DCT is available thus far[52–54]. To gain

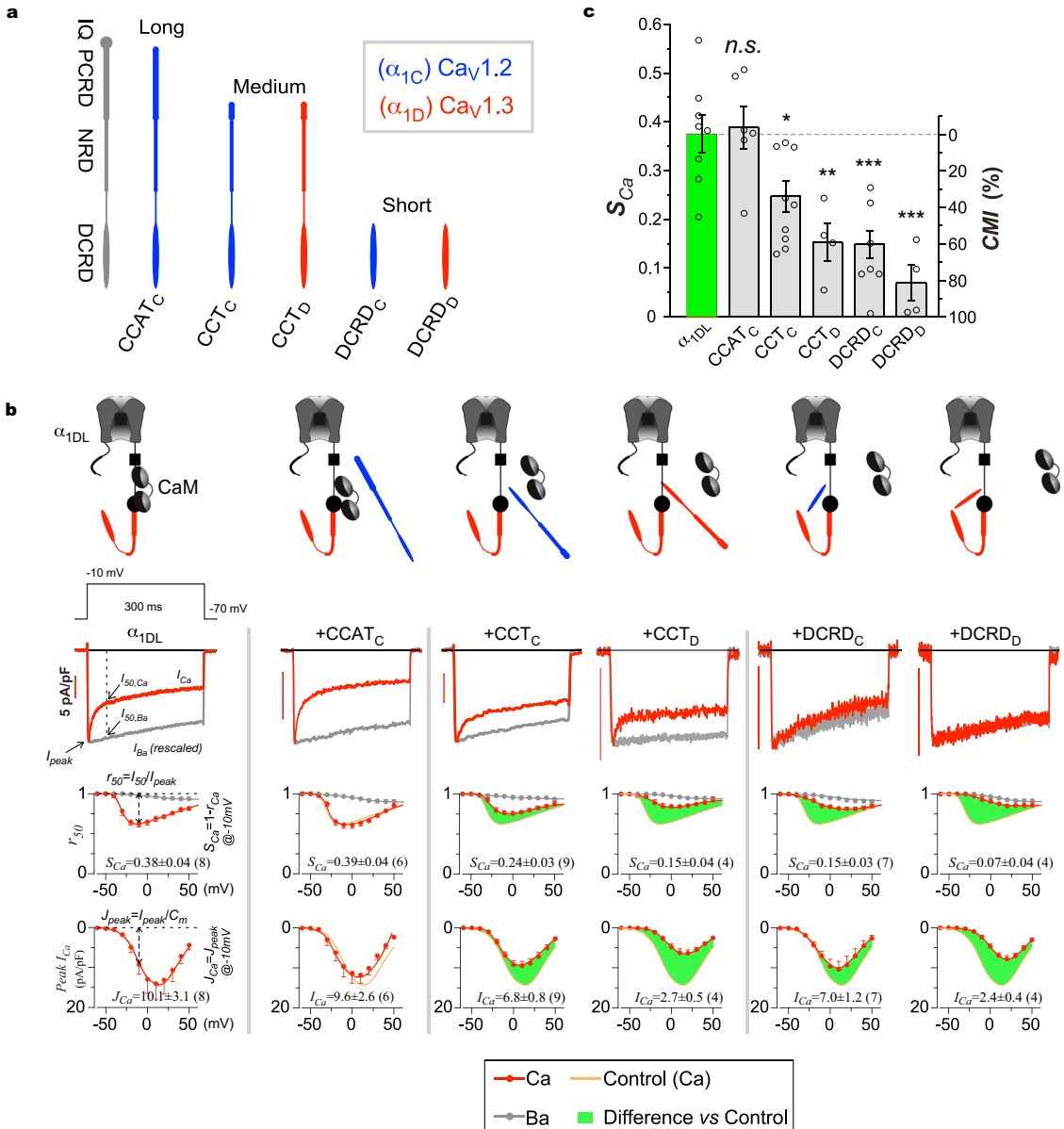

**Fig. 2 Inhibition of recombinant Ca$_V$1.3 channels by representative DCT peptides. a** Illustration of serial representative peptides encoded from DCT domains of Ca$_V$1.2 (blue) and Ca$_V$1.3 (red). The long-form peptides (~60 kDa) represented by CCAT$_C$ contain the complete set of motifs including PCRD, NRD and DCRD. The medium-form (~40 kDa) representatives CCT$_C$ and CCT$_D$ start from the very end of the PCRD till the end of the DCRD thus containing both the NRD and the DCRD. The short-form peptides (~15 kDa) are represented here by the peptides DCRD$_C$ and DCRD$_D$. **b** CMI effects of representative variants on Ca$_V$1.3 ($\alpha_{1DL}$) in HEK293 cells. As illustrated by the control group (left column), exemplary Ca$^{2+}$ current (trace with scale bar, red) and Ba$^{2+}$ current (rescaled to Ca$^{2+}$ current at the peaks, gray) were elicited by voltage step at −10 mV (top traces). The next two rows show the profiles of inactivation and activation respectively, with $r_{50}$ (ration between current amplitudes at 50 ms and the peak) for inactivation and $I_{peak}$ (Ca$^{2+}$ current) for activation across the full range of membrane potentials ($V$). Based on Ca$^{2+}$ currents at −10 mV, $S_{Ca}$ (in fraction, 1-$r_{50}$) and $I_{Ca}$ (in pA/pF, $I_{peak}$) serve as the major indices for inactivation and activation, respectively. Five representative peptides of CCAT$_C$, CCT$_C$, CCT$_D$, DCRD$_C$ and DCRD$_D$ (from left to right) as in (**a**) are compared with $\alpha_{1DL}$ control (the leftmost column) for their inactivation and activation profiles (lines in orange indicate the control). Green areas are to highlight peptide effects. **c** Statistical summary of the extent of Ca$^{2+}$-dependent inactivation ($S_{Ca}$) and peptide's CMI potency (*CMI* in percentage). To evaluate the attenuation of DCT peptide on inactivation ($S_{Ca}$), CMI potency is defined as the change in Ca$^{2+}$-dependent inactivation ($S_{Ca}$): ($S_{Ca,Control}$ − $S_{Ca,Peptide}$)/$S_{Ca,Control}$, which is equivalent to fractional change in apoCaM-bound channels ($\Delta F_{CaM}$) before ($F_{CaM}$) and after DCT peptide's competition. Thus, *CMI* essentially indicates what percentage of apoCaM-bound channels are converted to peptide-bound channels ($f_{Peptide}$). See Eqs. E2 and E3 in Methods for more details. One-way ANOVA followed by Dunnett for post hoc tests were used for (**c**): *$p < 0.05$; **$p < 0.01$; ***$p < 0.001$. Values are represented as mean ± SEM.

further insights into the mechanisms underlying DCT effects, we firstly focused on CCAT$_C$, unexpectedly exerting rather mild inhibition on Ca$_V$1.3 channels and cortical neurons (Fig. 3g). In contrast to CCAT$_C$, the shorter peptides of DCRD$_C$ and CCT$_C$ both encoded by Ca$_V$1.2 DCT have strong CMI, suggesting a self-

limiting mechanism within the longer CCAT$_C$. Moreover, Ca$_V$1.2 has been considered to have the same level of Ca$^{2+}$-dependent inactivation with or without its DCT domain[22,55], inconsistent with strong CMI of DCRD$_C$ in our experiments (Fig. 2). To resolve these discrepancies, we performed systematic analysis

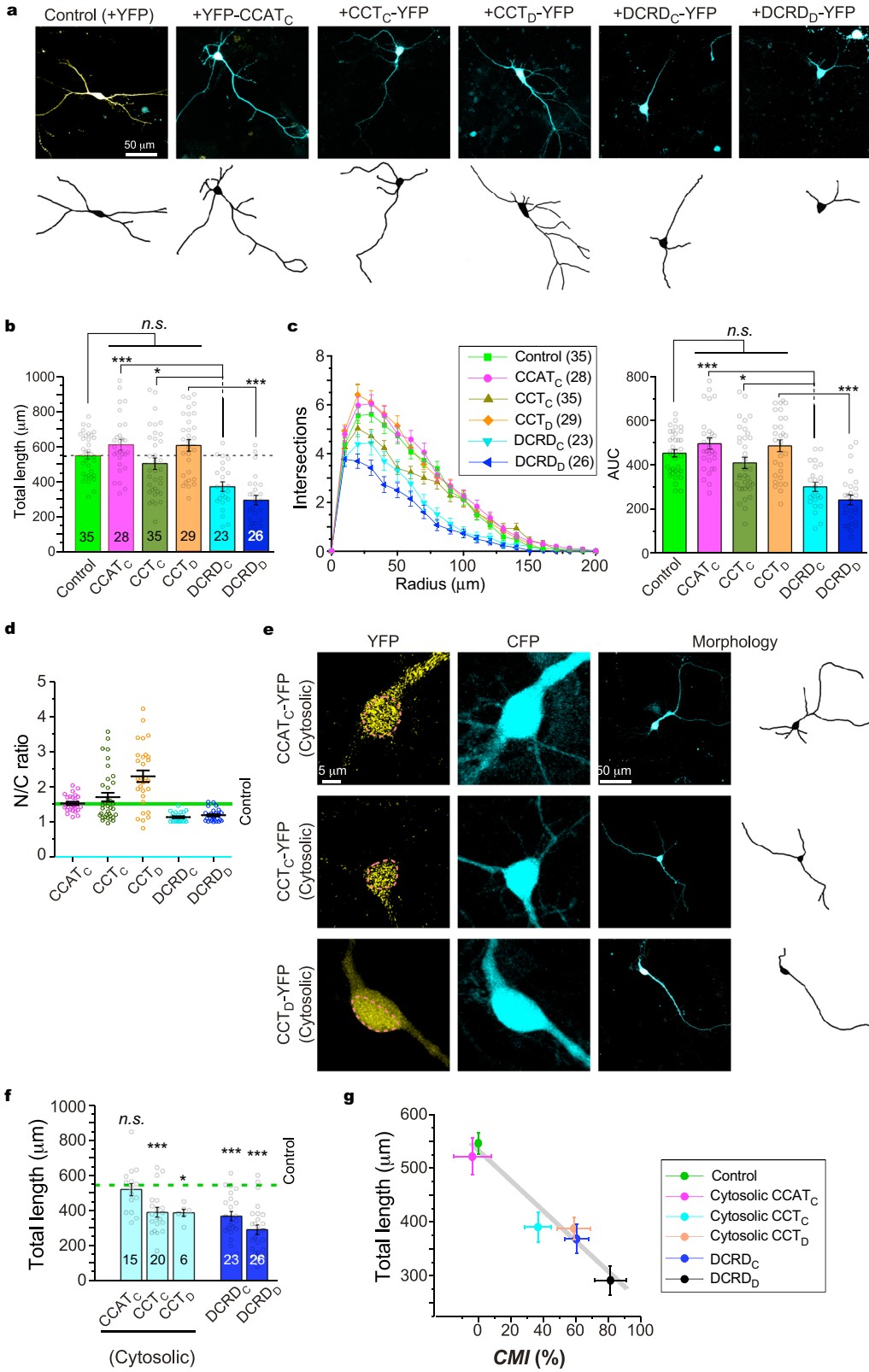

with the representative DCT peptide variants. By utilizing 2-hybrid 3-cube FRET (Förster resonance energy transfer), a quantitative imaging assay for protein-protein interactions in live cells[24,56], the capabilities of DCRD peptides to bind the channel were quantified by dose-dependent binding ($FR\text{-}D_{free}$) curves (Fig. 4a). Following the convention, we employed the (effective) dissociation equilibrium constant ($K_d$, units in fluorescence intensities through the donor cube) as the index of binding affinities. Utilizing CFP-tagged $DCRD_X$ peptides (X = S, D, C, and F representing $Ca_V1.1\text{-}1.4$) and YFP-tagged $preIQ_3\text{-}IQ_D\text{-}PCRD_D$ (CaM-binding motif of $Ca_V1.3$) as the FRET pairs, a series of binding curves were achieved by iterative fitting processes (Fig. 4a). Among a gradient of $K_d$ values from the four pairs of binding, $DCRD_F$ encoded by $Ca_V1.4$ resulted

**Fig. 3 Cytosolic DCT peptides inhibit neurite outgrowth of cortical neurons. a** Fluorescence images of cortical neurons (DIV-7) expressing CFP and DCT variants tagged with YFP. Representative confocal images (upper row, merged CFP in cyan with YFP in yellow) and corresponding neurite tracing (bottom row) are shown for YFP control, CCAT$_C$, CCT$_C$, CCT$_D$, DCRD$_C$ and DCRD$_D$, respectively. **b, c** Statistical summary of total neurite length (**b**) and Sholl analyses (**c**). **d** Statistical summary of N/C ratio for all the five groups of DCT peptides. Horizontal line indicates N/C ratio of YFP as the criteria (for DCT distribution) to assign each neuron to either the nuclear subgroup (N/C ratio>1.5) or the cytosolic subgroup (N/C ratio<1.5). **e** Representative confocal images, neurite tracings and detailed cytonuclear distribution are shown for neurons with cytosolic CCAT$_C$, CCT$_C$ and CCT$_D$, respectively. The envelope of the nucleus is highlighted by live-cell Hoechst 33342 staining (dotted lines). **f** Total length for the neurons from the cytosolic groups: peptides of CCAT$_C$, CCT$_C$ and CCT$_D$, and also the groups of DCRD$_C$ and DCRD$_D$ peptides. **g** A potential correlation between CMI values (adopted from Fig. 2c) and total neurite length ($R^2 = 0.93$). One-way ANOVA followed by Bonferroni or Dunnett for post hoc tests were used for (**b**, **c**) and (**f**), respectively (*$p < 0.05$; **$p < 0.01$; ***$p < 0.001$). Values are represented as mean ± SEM.

in the strongest affinity ($K_d = 1.8 \times 10^3$), followed by the peptides DCRD$_D$ ($K_d = 4.3 \times 10^3$), DCRD$_C$ ($K_d = 16.5 \times 10^3$) and DCRD$_S$ ($K_d = 29.0 \times 10^3$).

In parallel with FRET binding analyses, the whole-cell electrophysiology was performed for functional characterizations. DCRD$_X$ peptides were overexpressed with α$_{1D\Delta}$-PCRD$_D$, a channel variant producing ultra-strong Ca$^{2+}$-dependent inactivation (due to lacking the critical DCRD domain) thus providing an ample dynamic range to evaluate CMI effects. All of the four DCRD$_X$ peptides caused inhibitory effects of different potency on α$_{1D\Delta}$-PCRD$_D$ channels, illustrated by their inactivation ($S_{Ca}$) profiles (Fig. 4b). The classic ligand binding (Langmuir isotherm) equation (see Methods: Eq. E4) between inhibition potency *CMI* and binding affinity $K_d$ was utilized to describe the differential effects among the peptide variants (Fig. 4c). For the mutant DCRD$_F$_V/A, both peptide binding and channel inhibition were severely perturbed by the critical mutation (Supplementary Fig. 6), also agreeing well with the tuning curve of $K_d$-dependent *CMI* (Fig. 4c).

In parallel, the potencies of PCRD$_X$ (X = S, D, C and F) were examined with FRET pairs YFP-preIQ$_3$-IQ$_D$-PCRD$_X$ and CFP-DCRD$_F$. Similar to DCRD$_X$, $K_d$ values were obtained for the PCRD$_X$ peptides, unveiling the relative order of strength in binding (starting from the strongest): PCRD$_F$, PCRD$_D$, PCRD$_C$, and PCRD$_S$ (Fig. 4d). The difference in $K_d$ between PCRD$_C$ ($K_d = 11.3 \times 10^3$) and PCRD$_D$ ($K_d = 1.8 \times 10^3$) is even more pronounced than that between DCRD$_C$ and DCRD$_D$ (6.3-fold versus 3.8-fold), suggesting that the rather weak inhibition by DCT$_C$ (either as the intramolecular motif or the intermolecular peptide) is mainly attributed to its proximal domain PCRD$_C$. Such result is unexpected, since the PCRD motif has been considered to play a much lesser role (than the DCRD motif) in DCT effects. For instance, it has been reported that PCRD is not required for channels inhibition under the low CaM conditions whereas DCRD still remains indispensable to CMI[24]. For fair comparison, the pair of DCRD$_F$ and preIQ$_3$-IQ$_D$-PCRD$_D$ is taken as the principle reference (noted as PCRD$_D$/DCRD$_F$ or its abbreviation P$_D$/D$_F$) (Supplementary Table 1). All combinations of PCRD$_X$/DCRD$_X$ (abbreviated as P$_X$/D$_X$) are summarized to compare their $K_d$ values (Supplementary Fig. 7). Besides experimental values from FRET, $K_d$ for other P$_X$/D$_X$ combinations can also be roughly estimated according to the values assigned to P$_X$ and D$_X$. For validation purposes, FRET experiments were conducted for P$_C$/D$_C$ (Ca$_V$1.2) and P$_S$/D$_S$ (Ca$_V$1.1) (Supplementary Fig. 8), resulted in rather weak binding affinities ($K_d$), consistent with the predictions from P$_X$ and D$_X$ (Supplementary Fig. 7).

Similar to DCRD, the functional role of PCRD was also examined, but by co-expressing PCRD$_X$-DCRD$_F$ (i.e., P$_X$-D$_F$) with α$_{1D\Delta}$ channels. P$_S$-D$_F$ and P$_C$-D$_F$ peptides exhibited much weaker inhibition than P$_D$-D$_F$ and P$_F$-D$_F$, indicated by less changes in Ca$^{2+}$-dependent inactivation of α$_{1D\Delta}$ channels (Fig. 4e). The four peptides of P$_X$-D$_F$ comply with the same

tuning curve of *CMI-$K_d$* in Fig. 4c, but here with P$_X$ as the factor subject to variations (Fig. 4f). In addition, the relatively weak $K_d$ and *CMI* for P$_C$-D$_C$ (Supplementary Fig. 8) agree well with the tuning curve, as one additional validation for its applicability to P$_X$-D$_X$ peptides.

In summary, both PCRD and DCRD may underlie the distinct effects of DCT peptides across the Ca$_V$1 family[20,22,57,58]. Here, the importance of PCRD is unmasked. The ultra-weak CMI potency of DCT$_S$ or DCT$_C$ is mainly attributed to its PCRD domain, in that DCRD$_S$ and DCRD$_C$ are fully capable of strong CMI effects (Fig. 4b). In this context, for CCT$_D$, CCT$_C$ and CCAT$_C$ (here equivalent to DCRD$_D$, DCRD$_C$, and DCT$_C$), the potencies of CMI effects (from strong to weak inhibition) on Ca$_V$1 (represented by Ca$_V$1.3, Fig. 2c) are expected to be in the same order as their binding affinities: P$_D$/D$_D$, P$_D$/D$_C$, and P$_C$/D$_C$ (from strong to weak binding).

**Compound effects on α$_{1DL}$ channels by long DCT peptides.** Although the weak inhibition by the long peptides DCT$_C$ or CCAT$_C$ could be attributed to its PCRD$_C$, it is still unclear why Ca$_V$1.3 (the full-length channel containing intramolecular PCRD$_D$) is barely regulated by CCAT$_C$ peptides. To fully elucidate the mechanism underlying the weak CCAT$_C$, we decomposed its effects on α$_{1D\Delta}$-PCRD$_D$ channels into two scenarios (Fig. 4g). The first component (I) represents the combination of P$_D$ from the channel and D$_C$ from the peptide, which produces strong inhibitory effects. The second component (II) represents the combination of P$_C$ and D$_C$ both from the long peptide DCT$_C$ (equivalent to CCAT$_C$), which has rather weak *CMI* (Supplementary Fig. 9a, b). Overall, CMI potency of CCAT$_C$ on α$_{1D\Delta}$-PCRD$_D$ is expected to fall into the range defined by both components (I and II) corresponding to the upper- and lower-limit respectively. The compound effects of CCAT$_C$ resulted in weak CMI potency toward its lower limit, suggesting a dominant role of PCRD$_C$ in this particular scenario. For validation purposes, P$_C$-D$_F$ (PCRD$_C$ fused with DCRD$_F$) was constructed and applied as an artificial type of DCT peptides (Supplementary Fig. 9c, d). Similar to P$_C$-D$_C$, the effects of engineered P$_C$-D$_F$ on α$_{1D\Delta}$-P$_D$ could also be decomposed into two combinations, where PCRD$_C$ compromised the ultrastrong CMI of DCRD$_F$ and thus the overall CMI only reached an intermediate level.

Collecting the data from PCRD or DCRD variants (Fig. 4c, f), a tuning curve between CMI potency and binding affinity (*CMI-$K_d$*) has been established, applicable to a broad scope of channel and peptide variants (Fig. 4g). In principle, for any DCT peptide variants, of either native or engineered and either WT or mutant forms, when applied to Ca$_V$1.3 channels (supposedly to Ca$_V$1 channels in general), the potency of CMI quantitatively would correlate with the affinity between peptides and channels, which is also a measure of the competition (against apoCaM) introduced by DCT peptides. Based on the tuning curve, $K_d$ values for

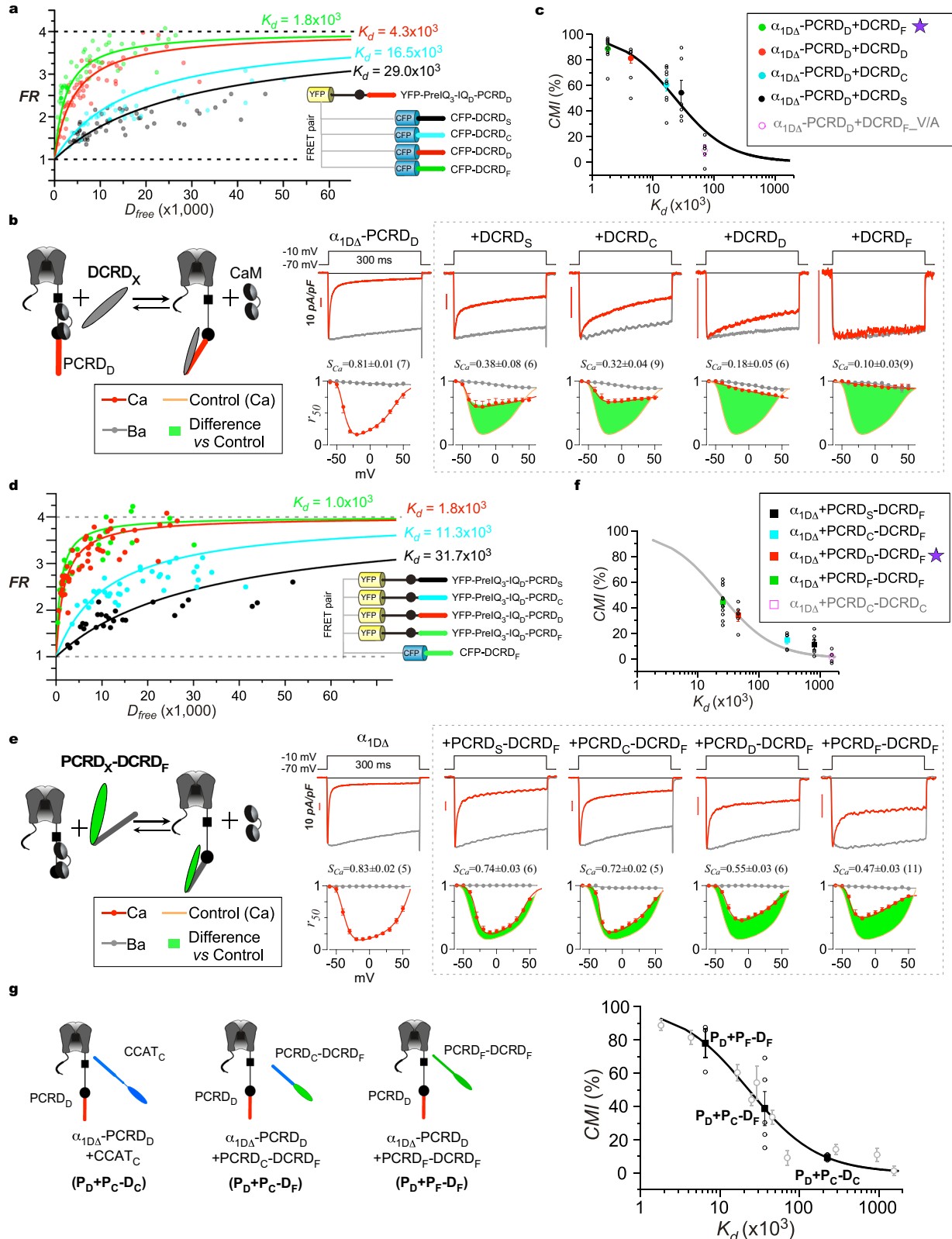

particular peptides can be estimated from their *CMI* values measured in electrophysiology, which has been demonstrated by P$_C$-D$_C$ (or CCAT$_C$) and P$_C$-D$_F$, and also by another long peptide P$_F$-D$_F$ (Supplementary Fig. 9e, f). Furthermore, CCAT$_S$[36], hypothetical CCAT$_D$ or CCT$_F$, and more other variants, can also be evaluated or predicated for the effects on Ca$_V$1 channels according to such unified tuning curve of *CMI*-$K_d$.

**Cytosol/nucleus-dependent effects of DCT peptides reconcile the discrepancy in neurons.** Our data thus far demonstrate that cytosolic DCT peptides negatively regulate neurite outgrowth, intrinsically tuned by *CMI* (channel inhibition) or $K_d$ (peptide binding) in a variant-dependent manner. The apparent contradictions regarding DCT effects (inhibitory versus facilitatory) may simply reflect the differential roles of peptides in the cytosol

**Fig. 4 Both PCRD and DCRD play important roles in CMI unveiled by DCT variants. a** Binding curves for the interactions between key channel motifs and DCRD peptides were quantified by 2-hybrid 3-cube FRET, for each pair between CFP-DCRD$_X$ from Ca$_V$1 (X = S, D, C and F) and YFP-preIQ$_3$-IQ$_D$-PCRD$_D$. $D_{free}$ and $FR$ represent free donor concentration and FRET ratio, respectively. The binding affinity $K_d$ for each pair was achieved by iterative fitting. In particular, the pair of PCDR$_D$ and DCRD$_F$ serves as the major reference for subsequent analyses. **b** Comparison of CMI potency among DCRD$_X$ peptides. As in the cartoon illustration of peptide CMI (top), the peptide DCRD$_X$ could coordinate with the IQ$_D$ and PCRD$_D$ motifs on α$_{1D\Delta}$-PCRD$_D$ to compete apoCaM off the channel. Ca$^{2+}$ trace exemplars and inactivation profiles are shown for α$_{1D\Delta}$-PCRD$_D$ alone or with different DCRD$_X$ isoforms. **c** Relationship between $K_d$ and $CMI$ for DCRD$_X$ peptides. Four peptides directly from Ca$_V$1.1-Ca$_V$1.4, plus one additional mutant peptide DCRD$_F$_V/A (Supplementary Fig. 6). The relationship between $K_d$ and $CMI$ for DCRD$_X$ peptides was fit by Eq. E4 (see Methods). **d** Similar to DCRD measurements (**a**), PCRD$_X$ peptides across Ca$_V$1 family (X = S, D, C, and F) were also quantified by FRET for the interactions between YFP-preIQ$_3$-IQ$_D$-PCRD$_X$ and CFP-DCRD$_F$. The two sets of measurements share the common pair PCRD$_D$ and DCRD$_F$ as the major index. See $K_d$ values of PCRD$_X$-DCRD$_X$ in Supplementary Fig. 7. **e** Comparison of CMI potency among PCRD$_X$-DCRD$_F$ peptides. As illustrated by the cartoon (top), PCRD$_X$-DCRD$_F$ could form the complex with IQ$_D$ of α$_{1D\Delta}$ to compete with apoCaM. In similar fashion, Ca$^{2+}$ trace exemplars and inactivation profiles are shown for α$_{1D\Delta}$ alone or with long-form peptides PCRD$_X$-DCRD$_F$. **f** Further support for the proposed $CMI$-$K_d$ correlation. Additional data points ($CMI$ and $K_d$) of PCRD$_X$-PCRD$_F$ (**e**) and PCRD$_C$-DCRD$_C$ (Supplementary Fig. 8) are superimposed onto the ligand-binding curve from (**c**). **g** The tuning curve of $K_d$-dependent $CMI$ with the summary of peptides. For compound effects of long-form DCT peptides on α$_{1D\Delta}$-PCRD$_D$, three more data points (black squares) were added onto the tuning curve, based on $CMI$ measurements and $K_d$ estimations for P$_D$+P$_C$-D$_C$ (or P$_D$+CCAT$_C$), P$_D$+P$_C$-D$_F$ and P$_D$+P$_F$-D$_F$ (Supplementary Fig. 9). Values are represented as mean ± SEM.

versus in the nucleus. In order to test this hypothesis, we first revisited the overexpressing CCT$_D$ which was widely distributed across the whole cell, featuring a broad range of N/C ratio values (Fig. 3d). Dual directional effects are evident: for the nuclear group CCT$_D$ (N) with N/C ratio>1.5, neurite outgrowth was promoted; in contrast, for neurons from the cytosolic group (N/C ratio<1.5) or CCT$_D$ (C), neurite outgrowth was significantly reduced similar to DCRD$_D$ (Fig. 5a–c). To confirm this result, the short tags of nuclear export signal (NES) and nuclear localization signal (NLS) were fused to the N-terminus of CCAT$_C$ or CCT$_C$. In doing so, NES-tagged CCAT$_C$ and CCT$_C$ were predominantly expressed in the cytosol (Fig. 5d, e). In comparison with the minor effects of NES-CCAT$_C$, neurite retractions were evidenced from NES-CCT$_C$ as indicated by shorter neurites (Fig. 5f) and reduced complexity (Fig. 5g), consistent with CMI-dependent inhibition of neuritogenesis we observed earlier (Fig. 3g). In contrast, NLS-tagged CCAT$_C$ and CCT$_C$ were constrained in the nucleus, presumably acting as neuritogenic transcription factors[19] (Fig. 5d–g). Indeed, both peptides led to the significant promotion of neurite outgrowth, whereas the longer peptide NLS-CCAT$_C$ was slightly more potent than the shorter NLS-CCT$_C$ (Fig. 5f), suggesting that the second TA (transcription activation) region (roughly overlapped with DCRD) may play a major role and the first TA region (largely overlapped with PCRD) would be relatively less significant (Supplementary Fig. 1). By revisiting Fig. 3, the actual effects on neurite outgrowth have been clarified to be highly dependent on subcellular localization of DCT peptides, which could be inhibitory when present in the cytosol by attenuating Ca$_V$1 activities and signals, or could be neuritogenic when localized in the nucleus as transcription factors (Fig. 5h). Furthermore, such opposing effects have been confirmed with more mature neurons (>DIV 15) of which neurite outgrowth was suppressed or facilitated by CCT$_D$ explicitly tagged with NES or NLS, respectively (Supplementary Fig. 10).

Cytosolic and nuclear DCT peptides are in direct opposition to each other (inhibition versus facilitation) in regulating neuritogenesis, as the plausible reason to account for the less-pronounced overall effects observed from overexpressing CCAT$_C$, CCT$_C$, and CCT$_D$ (Fig. 3a–c). On the one hand, nuclear DCT peptides are able to promote neuritogenesis; but on the other hand, cytosolic peptides have distinct CMI potency to induce differential levels of inhibition on neurite outgrowth. Is there any mechanism to regulate/maintain the potential balance between the opposing (cytosolic versus nuclear) effects? First, the spatial distribution was examined for DCT peptides in accordance to neurite outgrowth. As depicted by the scatter plots to correlate N/C ratio with total neurite length, the peptides CCAT$_C$, CCT$_C$

and CCT$_D$ (compared with CCT$_D$_V/A) spread across the cytosol/nucleus of each neuron (Fig. 5i). In contrast, the DCRD$_C$ and DCRD$_D$ peptides were exclusively constrained within the cytosol thus solely functioning as inhibitors of neurite outgrowth (Supplementary Fig. 11). Roughly at the same expression levels for whole cells, these peptides exhibited different patterns of subcellular distribution, as illustrated by the dynamic range between the minimum and maximum N/C ratios for each peptide. For CCAT$_C$, the rather narrow range of N/C ratio is consistent with its least localization in the nucleus. In comparison, CCT$_D$ appears to spread into the nucleus with a broader range of N/C ratio. We quantified the dynamic range of peptide distribution by the standard deviation (σ) of N/C ratio, which is closely correlated with CMI potency in the order of CCAT$_C$, CCT$_D$_V/A, CCT$_C$, and CCT$_D$ from weak to strong (Fig. 5j). Regarding the mutant peptide CCT$_D$_V/A, the single-residue V/A mutation in the DCRD domain significantly attenuated its inhibitory effects on channel gating (Supplementary Fig. 12a–c), and neurite outgrowth (Supplementary Fig. 12d–f). Notably, nuclear CCT$_D$_V/A still promoted neurite outgrowth, similar to WT CCT$_D$ in the nucleus. In agreement with the proposed σ-CMI correlation, CCT$_D$_V/A is less distributed in the nucleus compared with WT CCT$_D$ (Fig. 5j); also, the clear differences between CCT$_D$_V/A and WT CCT$_D$ resemble CCAT$_C$ versus CCT$_C$. One potential explanation could be that the distributions and effects of DCT peptides are subject to certain autonomous regulations in neurons, presumably by way of Ca$_V$1/Ca$^{2+}$ influx and Ca$^{2+}$-sensitive NRD (the nucleus retention domain contained in the long or medium peptides but not in the short peptides, Supplementary Fig. 1).

**Downregulations of neuritogenesis signaling by Ca$_V$1.3-encoded peptides.** The localization-dependent regulation of neuritogenesis reconciles the opposing roles of exogenous DCT peptides in neurons. Before we proceeded further with its potential Ca$^{2+}$/Ca$_V$1 dependence, we examined the key signals that are involved. For Ca$_V$1.3-encoded peptides, CCT$_D$ should serve as one representative form, considering that all the key domains such as NRD are included and there is no issue of weak PCRD as in CCAT$_C$ (Supplementary Fig. 1). As expected, the exogenous peptides of NES-CCT$_D$ demonstrated that the cytosolic peptides suppressed pCREB in direct contrast to the nuclear peptides of NLS-CCT$_D$ (Fig. 6a). c-Fos, a hallmark gene of DCT effects (Fig. 1d), was also examined here for its expression driven by Ca$_V$1/pCREB. Resembling pCREB, c-Fos expression was significantly reduced by NES-tagged but not by NLS-tagged peptides

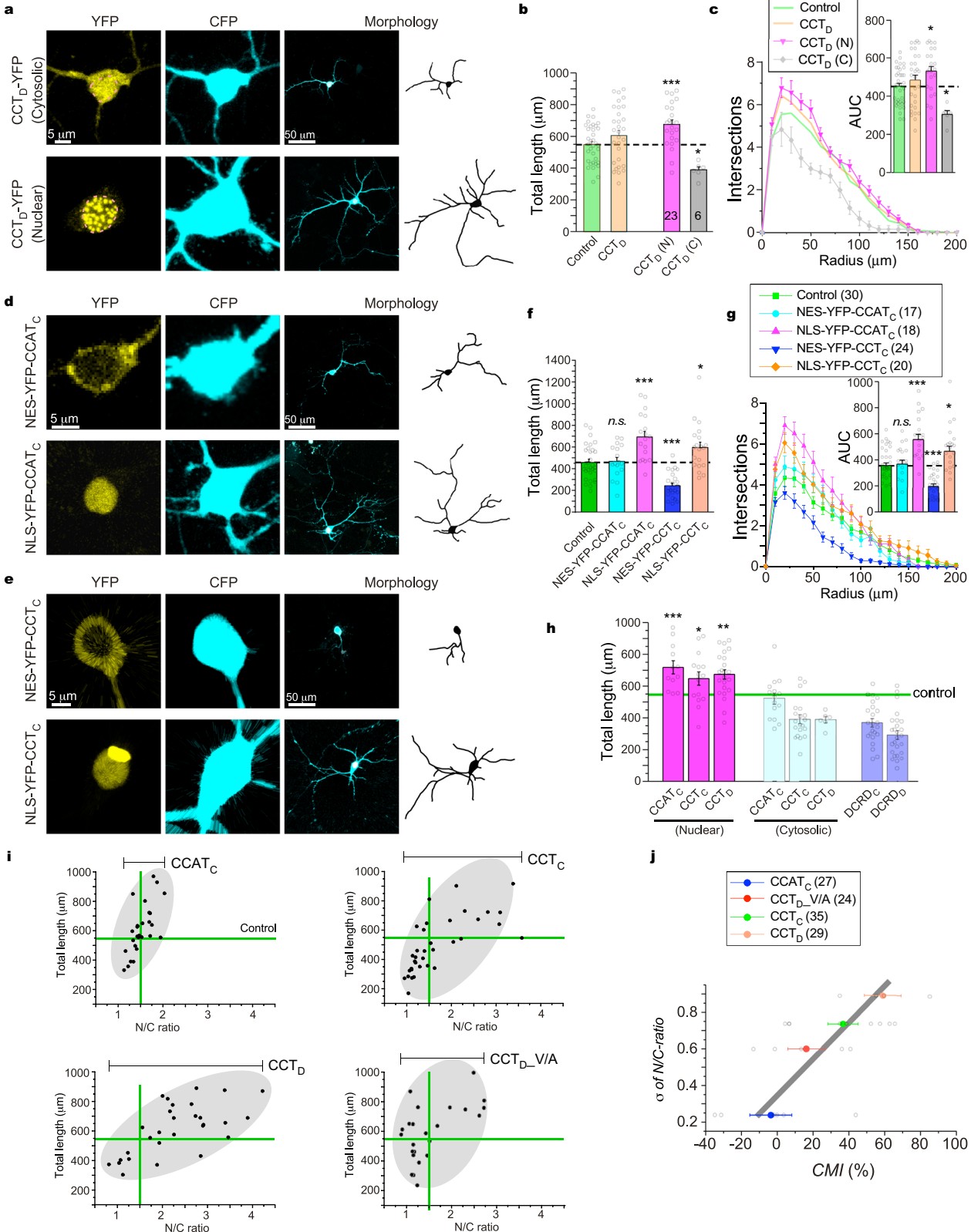

(Fig. 6b), consistent with that nuclear DCT peptides are able to directly serve as transcription factors to promote expression of neuritogenic genes. Next, we attempted to explore whether $Ca_V1.3$-encoded peptides would play an endogenous role in cortical neurons. By anti-$CT_D$ immunostaining, the subcellular distribution of $CT_D$ peptides exhibited a potential correlation with neurite outgrowth (Supplementary Fig. 13a, b). In agreement

with exogenous peptide effects, endogenous $CT_D$ in the cytosol appeared to suppress pCREB signaling (Supplementary Fig. 13c). The antibodies of anti-$CT_D$ and anti-$CT_C$ resulted into western-blot bands of differential sizes (Supplementary Fig. 14a, b and Supplementary Fig. 17a, b), supporting antibody specificity with no cross-reactivity (Supplementary Figs. 14c, 17c). Meanwhile, a bicistronic mechanism[19,43] may underlie the peptide production

**Fig. 5 Opposite effects of cytosolic versus nuclear DCT peptides on neurite outgrowth. a** Cortical neurons with CCT_D fragments are grouped into two categories: nuclear *versus* cytosolic, i.e., CCT_D (N) and CCT_D (C) by the same criteria of N/C ratio as in Fig. 3 (CCT_D fluorescence with N/C ratio >1.5 for the nuclear group; and N/C ratio <1.5 for the cytosolic group). **b, c** Total neurite length (**b**) and Sholl analyses (**c**) for neuron groups of Control, total CCT_D, CCT_D (C) and CCT_D (N). Original data for Control and total CCT_D groups are adopted from Fig. 3b-e NES and NLS were fused to N-terminus of YFP-CCAT_C (**d**) or YFP-CCT_C (**e**) to constrain the distribution of DCT peptides within the cytosol or the nucleus, respectively. **f, g** Total neurite length (**f**) and Sholl analyses (**g**) for the groups of Control, NES-YFP-CCAT_C versus NLS-YFP-CCAT_C, and NES-YFP-CCT_C versus NLS-YFP-CCT_C. **h** Total neurite lengths for neurons overexpressing CCAT_C, CCT_C and CCT_D peptides are summarized and compared among nuclear versus cytosolic subgroups, together with DCRD_C and DCRD_D groups (adopted from Fig. 3f). **i** Cyto-nuclear distribution (indexed by N/C ratio) of DCT peptides in correlation with the total neurite length. Horizontal and vertical lines in green represent the control group (YFP expressed in cortical neurons). The range of N/C ratio is indicated on top of the scatter plot for each peptide group of CCAT_C, CCT_C, CCT_D, or the mutant CCT_D_V/A. The grey shades for peptide variants are to illustrate the potential correlation between neurite length and N/C ratio. **j** Correlation between CMI strength and cyto-nuclear distribution. For four peptide variants of CCAT_C, CCT_C, CCT_D and CCT_D_V/A, standard deviations of N/C ratio values are represented by $\sigma$ (*N/C ratio*) to quantify spatial dynamics for each peptide variant. *CMI* and $\sigma$ are highly correlated (linear fit, $R^2 = 0.90$). One-way ANOVA followed by Dunnett for post hoc tests were used for (**b**, **c**), and (**f**–**h**) (*$p < 0.05$; **$p < 0.01$; ***$p < 0.001$). Values are represented as mean ± SEM.

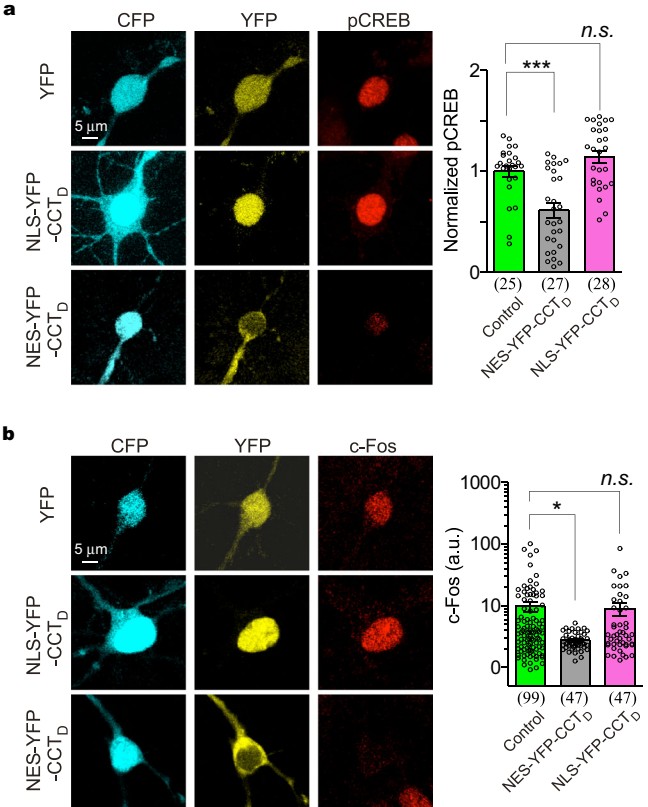

**Fig. 6 Effects of Ca_V1.3-encoded peptides on neuritogenesis signaling in cortical neurons. a**, **b** Inhibitory effects on pCREB and c-Fos by cytosolic peptides of exogenous CCT_D. Confocal fluorescence images of cortical neurons expressing YFP, NLS-YFP-CCT_D or NES-YFP-CCT_D. YFP indicates CCT_D distribution, and CFP illustrates cell bodies, respectively. Normalized pCREB (**a**) and c-Fos fluorescence (**b**) in the nuclei are summarized and compared. One-way ANOVA followed by Dunnett for post hoc test (**a**), and Kruskal-Wallis and Dunn's non-parametric test (**b**, non-normal distribution, checked by D'Agostino & Pearson omnibus normality test) (*$p < 0.05$; ***$p < 0.001$; *n.s.* denotes not significant, $p > 0.05$). Values are represented as mean ± SEM.

from the full-length $\alpha_{1CL}$ or $\alpha_{1DL}$ transfected into HEK cells (Supplementary Figs. 14d–f, Fig. 17d–f).

In summary, Ca_V1.3-encoded DCT peptides, if exogenously expressed in the cytosol, suppressed Ca_V1/CREB-mediated neuritogenesis, which may represent the role of endogenous DCT peptides in cortical neurons awaiting future investigations.

## Ca_V1/Ca$^{2+}$ influx and CMI are critical to peptide distributions in neurons.

We discovered that subcellular distributions of DCT peptides are subject to autoregulation in accordance with peptide *CMI*, potentially through DCT inhibition of Ca_V1/Ca$^{2+}$ influx (Fig. 5i, j). In support, nuclear export of DCT peptides is enhanced by intracellular Ca$^{2+}$ rise as previously reported[19,40]. Ca$^{2+}$ influx, via Ca_V1 channels in particular, is critical to subcellular distributions (or cytosol-nucleus translocation) of DCT peptides, which was confirmed in this work (Supplementary Fig. 15). When Ca_V1 activities under basal conditions (5 mM [K$^+$]_o) were blocked by 50 μM nifedipine (DHP derivative), the tendency of CCT_D to translocate into the nucleus was significantly enhanced. In addition, N/C ratio of CCT_D returned back to the control level when DHP-insensitive channels ($\alpha_{1DL}$ DHP$^-$) were employed instead (Fig. 7a, b), supporting the unique importance of Ca_V1 in cytosol-nucleus translocation of CCT_D. In cortical neurons (supplied with 50 μM nifedipine) where endogenous Ca_V1 channels were replaced with DHP-insensitive channels, we further investigated whether different DCT motifs contained within the channels (through differential overall *CMI*) could produce any effect on peptide translocation. Four DHP-resistant Ca_V1.3 variants, in the order of $\alpha_{1D\Delta}$, $\alpha_{1DL}$_V/A, $\alpha_{1DL}$ and $\alpha_{1D\Delta}$-DCT_F, exhibited increasingly stronger DCT effects, i.e., leading to weaker activation/inactivation, and less fraction of apoCaM-bound channels (Fig. 7c). In this context, covalently-linked (intramolecular) DCT motifs comply with the same principle as standalone (intermolecular) DCT peptides to induce inhibitory effects (CMI), both able to elevate DCT-bound fraction (and reduce apoCaM-bound fraction) of channels, thus inhibiting Ca$^{2+}$ influx and Ca$^{2+}$/Ca_V1 signaling. Since $\alpha_{1D\Delta}$-DCT_F (DHP$^-$) channels would have the largest DCT-bound fraction ($f_{DCT}$) when expressing in neurons, more CCT_D peptides translocated into the nucleus and exhibited the largest variation in peptide's localization ($\sigma$ of N/C ratio) (Fig. 7d). In contrast, for neurons expressing the $\alpha_{1D\Delta}$ DHP$^-$ channels (the weakest *CMI* due to completely lacking DCT), the least nuclear retention of CCT_D (the smallest $\sigma$) was observed. This is presumably due to the fact that Ca$^{2+}$ influx via Ca_V1 channels and the related Ca$^{2+}$-sensitive nuclear export of CCT_D would be the most pronounced for $\alpha_{1D\Delta}$ DHP$^-$ among the four variants.

Effective CMI (incorporating both intra- and inter-molecular CMI) could be tuned by a series of standalone DCT peptides (acting on the same channel Ca_V1.3, Fig. 5j), or by replacing the full-length Ca_V1.3 with various DCT motifs (modulated by the same peptide CCT_D, Fig. 7d). Taken together, a tight correlation has been unveiled between DCT-peptide localization and Ca_V1-channel activities, where the central factor is overall DCT inhibition of Ca_V1 (effective CMI). To this point, Ca_V1-encoded DCT peptides comply with a type of self-regulatory scheme that

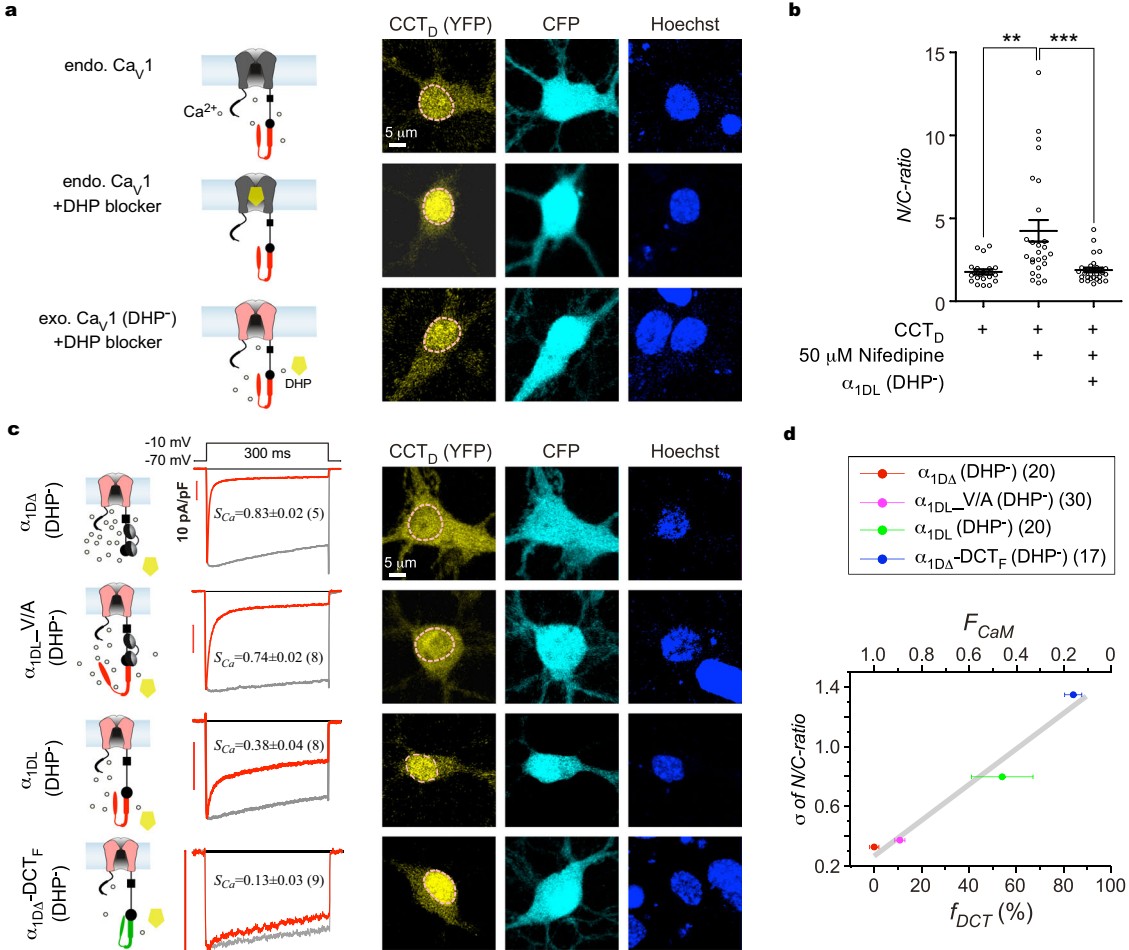

**Fig. 7 Unique importance of Ca$_V$1 to subcellular DCT distributions in neurons unveiled by CMI. a**, **b** Effects of dihydropyridine (DHP). As shown in the cartoon (**a**, left), endogenous Ca$_V$1 channels in cortical neurons mediate Ca$^{2+}$ influx (upper); DHP (50 μM nifedipine) specifically blocked cortical Ca$_V$1 channels thus reducing Ca$^{2+}$ influx (middle); and overexpression of α$_{1DL}$ (DHP$^-$) channels (mutant Ca$_V$1.3 insensitive to DHP) rescued Ca$^{2+}$ influx in the presence of 50 μM nifedipine (bottom). Exemplary fluorescence confocal images through three channels: YFP (CCT$_D$ distribution), CFP (cell contour) and Hoechst (nuclear envelop) (**a**, right). N/C ratios of individual neurons from the three groups: control, DHP, and α$_{1DL}$ (DHP$^-$) with DHP (**b**). **c** Ca$_V$1 variants containing DCT of different potency regulated cytosolic-nuclear distribution of CCT$_D$. In addition to CCT$_D$, cortical neurons were overexpressed with one of the four DHP-resistant Ca$_V$1.3 variants: α$_{1DΔ}$ (lacking DCT), α$_{1DL}$_V/A (with the key mutation of Valine to Alanine at DCRD), α$_{1DL}$ (control), and α$_{1DΔ}$-DCT$_F$ (chimera with ultra-strong DCT). These Ca$_V$1.3 variants were characterized by exemplar current traces from HEK293 cells expressing the variants illustrated in the cartoons. Cortical neurons were treated with 50 μM nifedipine to block endogenous Ca$_V$1 channels while sparing the exogenous DHP$^-$-Ca$_V$1.3 channels. Confocal images depict cytosolic-nuclear localizations of CCT$_D$ peptides in a similar fashion to (**a**). **d** Correlation between DCT inhibition of each Ca$_V$1.3 variant and spatial dynamics for CCT$_D$ peptides in cortical neurons. DCT/CMI potency (DCT-bound fraction $f_{DCT}$) here is directly linked to inactivation (fraction of apoCaM-bound channels, $F_{CaM}$): $f_{DCT} = 1 - F_{CaM}$ (Eq. E5 in Methods). DCT/CMI potency and standard deviation of CCT$_D$ distribution (σ) exhibited a tight correlation (linear fit, $R^2 = 0.95$). One-way ANOVA followed by Bonferroni for post hoc tests were used for (**b**) (**$p < 0.01$; ***$p < 0.001$). Data are represented as mean ± SEM.

DCT distributions are adjusted between the cytosol versus the nucleus through DCT inhibition of Ca$_V$1 channels.

## Discussion

In this study, we systematically examined a series of DCT-encoded peptides across the Ca$_V$1 family; and by focusing on the representative Ca$_V$1.3 in cortical neurons, we unveiled that DCT peptides through CMI inhibit the signaling cascade from Ca$_V$1 to neuritogenesis. One determining factor of the overall CMI effects is the DCT affinity with Ca$_V$1, contributed by both PCRD and DCRD segments of the peptide-channel complex. In parallel, the reduction of Ca$^{2+}$ influx by cytosolic DCT is in favor of nuclear localization of DCT acting as neurotrophic transcription factors. In all, Ca$_V$1 channel activities, Ca$_V$1 channel-mediated signaling to the nucleus, gene transcription related to neuritogenesis, and Ca$_V$1/Ca$^{2+}$-

sensitive nuclear export of DCT are all downregulated by cytosolic DCT peptides that bind and inhibit Ca$_V$1 channels (Fig. 8).

In this study, our major strategy was to utilize a series of representative DCT peptides covering the major variants across the Ca$_V$1 family. The central hypothesis has been that the DCT peptides with apparent distinctions share the same principles: the capabilities to downregulate Ca$_V$1 activity-dependent neuritogenesis via the interactions between DCT peptides and Ca$_V$1 channels. On the other hand, it would be unrealistic to exclusively examine all the signals or events along the whole pathway. Instead of multiple checkpoints with less rigor, our resolution was to focus on the key signals or indices, e.g., Ca$_V$1 gating or neurite morphology, but taking advantage of multiple peptides with a gradient of binding affinities and neuronal effects for quantitative consolidations.

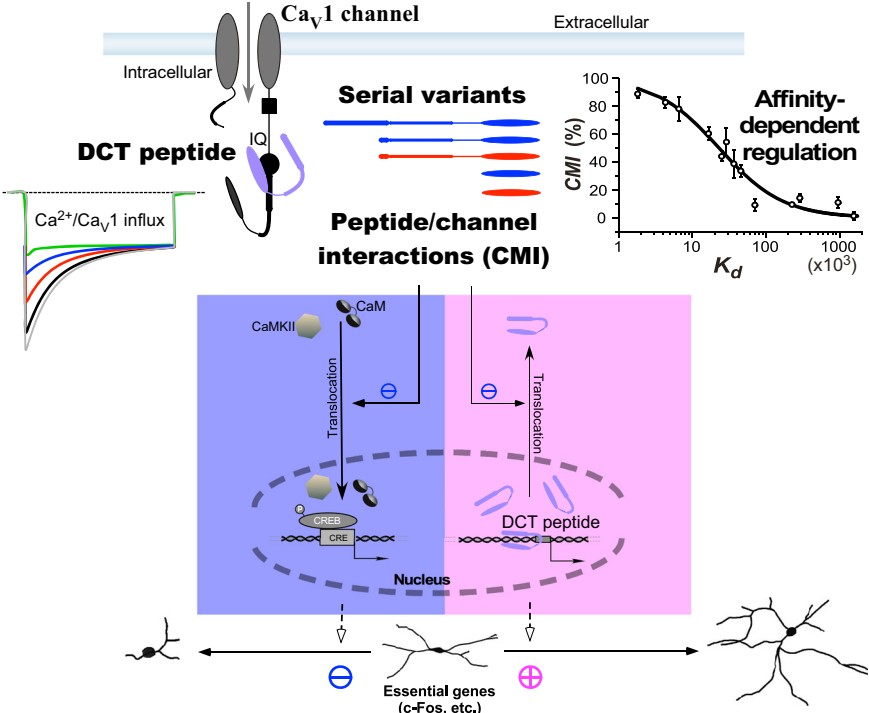

**Fig. 8 Cytosolic DCT peptides regulate $Ca_V1$ channel- and nuclear DCT-mediated neuritogenesis.** Regardless of the distinctions in mechanisms of production, molecular compositions, or modulatory effects, diverse peptides have been unified into a tuning curve of the central principle: DCT/$Ca_V1$ affinity-dependent inhibition of the channel activity-neuritogenesis coupling, which has been demonstrated by representative $Ca_V1.3$ channels in cortical neurons. A series of DCT peptide variants (color-coded to illustrate the difference in origin), by interacting with $Ca_V1$ channels, inhibit $Ca^{2+}$ influx (color-coded to illustrate the difference in potency) and regulate $Ca_V1$ signaling to the nucleus and gene expression, all quantitatively in accordance with peptide/channel affinity ($CMI-K_d$ relationship). For $Ca_V1$ channel bound with CaM at the preIQ-IQ domain, cytosolic DCT peptides compete against CaM to form peptide-channel complex. The capability of DCT competition is quantified as CMI potency to represent the fraction of channels being switched from CaM-bound to DCT-bound. DCT-channel affinities or CMI potencies are drastically different among serial peptide variants including those endogenous to native cells, varying in size (long, medium or short) and/or origin ($Ca_V1.1-1.4$). In close correlation with CMI potency, DCT peptides inhibit $Ca_V1$ gating and $Ca^{2+}$ influx, reduce nuclear translocation of key signaling molecules CaM (and CaMKII), and attenuate $Ca_V1$-mediated transcription (e.g., pCREB) and expression of essential genes (e.g., c-Fos), eventually leading to inhibition of neuritogenesis (in the blue shade). As collateral effects (in the pink shade), CMI also downregulates nuclear export of DCT peptides, and thus facilitates gene transcription directly mediated by nuclear DCT peptides, leading to promotion of neurite arborization and extension. In summary, we discover cytosolic DCT inhibition of the $Ca_V1$ activity-neuritogenesis coupling, which is in direct opposition to neurotrophic signaling of nuclear DCT.

Although some $Ca_V1$ inhibitors such as dihydropyridine (DHP) do exert inhibitory effects on $Ca_V1$-dependent signaling and neuritogenesis[10,11], it is still a difficult task to identify effective $Ca_V1$ inhibitors for $Ca_V1$-dependent neuritogenesis. First, the linkage from $Ca^{2+}$ influx to downstream signaling is not guaranteed, i.e., the potential decoupling between $Ca_V1$ channels and gene expressions, known as flux independence. In fact, the channel pore blocker $Cd^{2+}$ and the gating blocker nimodipine may behave very differently in their effects on pCREB signals[59]. Also, instead of neuritogenic effects, $Ca_V1$ agonist Bay-K-8644 causes neural toxicity[60,61]. In Timothy syndrome, gain-of-function $Ca_V1.2$ mutations that promote $Ca^{2+}$ influx cause neural damages due to ectopic activation of retractive signals[17]. Therefore, careful experiments and analyses are required to link modulation of $Ca_V1$ to neuritogenesis. Second, multiple factors besides $Ca^{2+}$ influx should work synergistically to ensure the complete signaling cascade. Less noticeable factors, e.g., voltage-dependent conformational changes of α or β channel subunits, may also play important roles to ensure proper signaling from $Ca_V1$ to the nucleus[59,62]. Channel inhibitors which only reduce the $Ca^{2+}$ influx may not attenuate neuritogenesis as effectively as expected. In this context, we have unveiled a class of $Ca_V1$-encoded peptide inhibitors endogenously present in neurons that effectively and consistently downregulate $Ca_V1$-dependent neuritogenesis, presumably by stripping apoCaM from the IQ domain of the channel.

The effects of DCT on channel gating appear to be divergent among $Ca_V1$ family members before this work. Whether CCTs could affect $Ca_V1.1$'s functions has been debated, perhaps due to different cellular environments and/or different truncation sites in these studies[26,57]. Moreover, for $Ca_V1.2$ channels, it has been reported that $DCT_C$ attenuates channel activation but does not affect $Ca^{2+}$-dependent inactivation[22], inconsistent with attenuation of inactivation evidenced from other reports[24,27]. In contrast, $DCT_D$ causes strong attenuation concurrently on both activation and inactivation, as the resolution of the contradictory effects on $Ca^{2+}$ influx; and $DCT_F$ and $DCT_D$ resemble each other except that $DCT_F$ inhibition is of even higher potency[20,21,24]. Here, we have provided a tuning scheme of CMI unified across $Ca_V1.1-1.4$ DCT (Fig. 8), demonstrated by representative $Ca_V1.3$ channels, which is expandable onto $Ca_V1.2$ (Supplementary Fig. 16) and other $Ca_V1$ channels. In particular, $DCRD_S$ and $DCRD_C$ are actually able to exert inhibition (CMI) of substantial potency, as opposed to previous observations or estimations although relatively less potent than $DCRD_D$ and $DCRD_F$. Importantly, we have clarified that the existing discrepancy in CMI potency among the DCT variants is critically dependent on the differences in PCRD isoforms. The weak effects of $DCT_S$ and $DCT_C$ are mainly

attributed to PCRD$_S$ or PCRD$_C$. Future structure-function analyses are needed to identify key PCRD residues and related mechanisms in detail. Two arginine residues reported earlier (R1696 and R1697 of Ca$_V$1.2) serve as the potential candidates on PCRD[22]. A few residues away from the above sites, i.e., S1575/T1579 or S1700/T1704 on Ca$_V$1.1 and Ca$_V$1.2 respectively, may provide some additional clues[63,64]. Our tuning curves could make the predictions for *CMI* and/or $K_d$ of diverse DCT peptides in principle (Fig. 4g and Supplementary Fig. 7). For example, regarding CCT$_S$ peptides generated by cleavage of Ca$_V$1.1 in skeletal muscle[36], its CMI potency by estimations would be moderate if acting on Ca$_V$1.3 (PCRD$_D$), and weak on Ca$_V$1.1 (PCRD$_S$) according to *CMI-$K_d$* relationship. Hence, it is unlikely that (cytosolic) CCT$_S$ could cause any strong inhibition of Ca$^{2+}$ influx via Ca$_V$1.1 in smooth muscle cells, which may help elucidate the existing arguments[36,57]. Despite important progress in Ca$_V$1 structures, none of these structures has acquired atomic details on DCT[52–54], which are foundational to understand DCT functions[20,24]. Our data here provide both the properties in common and the critical differences among DCT variants. FRET binding and electrophysiology data suggest that DCT is subject to the tight competition with CaM before the channel permits Ca$^{2+}$, based on which we postulate that DCT peptides may acquire the (apo)CaM-like structures. In this context, DCRD and PCRD may mimic the C- and N-lobe of CaM respectively. First, apoCaM usually binds the target (such as the IQ domain of neuromodulin or neurogranin) with its C-lobe[65]; similarly, DCRD as one of the two helical subdomains plays a dominant role in the DCT/apoCaM competition (for binding the IQ domain of Ca$_V$1). In comparison, PCRD appears to be assistive, e.g., to properly anchor DCRD in the close vicinity (of the channel). Second, the interactions between the CaM-binding motif and EF-hand containing CaM-like proteins are mainly mediated by charged and aromatic/hydrophobic residues[65,66], which are also similarly enriched in DCT. The functional and structural details of CMI/DCT would advance our understanding about how apoCaM binds Ca$_V$1 and promotes its functions[23,67].

In this work, a neurotrophic role has been confirmed for DCT peptides localized in the nucleus; meanwhile, cytosolic DCT peptides inhibit neurite outgrowth. Potentially, the overall effects may constitute a homeostatic balance sustained by two signaling opponents (cytosolic versus nuclear) in neurite morphogenesis (Fig. 3b, Fig. 5h and Supplementary Fig. 13b). DCT/CMI inhibits the Ca$_V$1 activity-neuritogenesis coupling represented by the following crucial signals or events: translocation of CaM/CaMKII from cytosol to nucleus, phosphorylation of CREB, and transcription and expression of hallmark genes (e.g., c-Fos)[11,14,49,68,69]. Higher CMI potency leads to less Ca$^{2+}$ influx via Ca$_V$1, eventually causing more pronounced retraction of neurites. On the other hand, less Ca$^{2+}$ influx resulted from potent DCT peptides tends to cause more nuclear retention due to Ca$^{2+}$-dependent nuclear export of DCT peptides. Nuclear DCT peptides as transcription factors drive the expression of a spectrum of neurotrophic genes[19]. Notably, DCT in the nucleus is autonomously regulated by DCT in the cytosol through its inhibition on Ca$_V$1. For example, under our experimental conditions of basal (channel/neuron) activities, CCAT$_C$ exhibits ultraweak inhibition of Ca$_V$1 but with a larger fraction of cytosolic distribution, opposed to nuclear DCT peptides of a relatively smaller fraction. CCT$_D$ is much more potent in Ca$_V$1 inhibition but a relatively small cytosolic fraction (in opposition to nuclear CCT$_D$ of a larger fraction), hence the tendencies of retraction/maintenance versus outgrowth could be substantially balanced out. For CCAT$_C$ or CCT$_D$, similar autoregulatory mechanisms may account for the rather mild CMI effects on neuritogenesis (Fig. 3b). We postulate that CMI regulation of endogenous DCT distribution would be employed to maintain a delicate balance for neuritogenesis, the setting points of which may vary with developmental stages, external stimuli or cues, and actual peptide variants, in addition to expression levels. Such tuning scheme of homeostasis is expected to generally apply to other types of neurons. Besides Ca$_V$1.3 and Ca$_V$1.3-encoded DCT peptides (e.g., CCT$_D$ and CT$_D$) as the focus of this work, Ca$_V$1.2 and Ca$_V$1.2 DCT peptides (e.g., CCAT$_C$ and CCT$_C$) are widely expressed in the brain[3,43,44]. Compared with Ca$_V$1.3, Ca$_V$1.2 channels should have less potent CMI effects due to the weak PCRD$_C$ motif, as another reason for this work to focus on Ca$_V$1.3. In cerebellar granule cells, CCAT$_C$ peptides serve as nuclear transcription factors that promote neurite ourgrowth[19]. CCAT$_C$ has also been evidenced in cell nuclei of the cerebellum and thalamus in embryonic brain, exporting to the cytosol along with aging and development[41]. CCT$_C$ by proteolysis has been found in hippocampus neurons[37], and hippocampal Ca$_V$1 channels are required for normal neurogenesis[45].

In addition to Ca$_V$1, Ca$_V$2.1 and Ca$_V$3.2 could encode peptides targeting the nucleus to regulate gene transcription by the bicistronic mechanism, which might be conserved across the superfamily of voltage-gated Ca$^{2+}$ channels[41–43,70]. C-terminal fragments of α$_{1A}$ act as transcription factors to promote neuronal development[42,43], resembling the effects of Ca$_V$1 DCT in the nucleus. Unlike the dual roles of Ca$_V$1 DCT peptides in this study, Ca$_V$2 CT has not been found to have any effect on channel gating[20,71]. Besides the autoregulatory scheme proposed here (Fig. 8), some other forms of feedbacks may also exist, e.g., Ca$_V$1.2-encoded peptides could reduce the transcription of Ca$_V$1.2 gene when located in the nucleus of cardiac myocytes[38,72]. Although no such downregulation has been evidenced from Ca$_V$1.3 in either recombinant systems or cortical neurons, more direct examinations are necessary to confirm the actual expression of functional Ca$_V$1.3 channels in different scenarios. Ca$_V$1-encoded polypeptides exhibit Ca$^{2+}$-dependent nuclear export, with the aid of its nucleus retention domain NRD[19], also supported by the distinctions between the short and long/medium peptides demonstrated in this study (Fig. 5h, i and Supplementary Fig. 11). Ca$^{2+}$-dependent DCT translocation is particularly sensitive to Ca$^{2+}$ via Ca$_V$1 (Fig. 7), while the exact mechanisms of DCT translocation are still awaiting future investigations[19].

The specificity of the antibodies is critical given that non-specific activities of Cav antibodies are not uncommon, in part due to the low expression levels of these membrane channels compared to other proteins. In the future, rigorous validations, e.g., with Ca$_V$1.3 or Ca$_V$1.2 knock-out (ideally conditional knock-out) neurons and/or the control of blocking peptides, are expected to confirm the western-blot and immunostaining data. A set of consistent data would strengthen the conclusions, if multiple approaches including electrophysiology, biochemistry and imaging could be combined together. For future work, additional methods/tools are expected, such as Ca$_V$1.3 antibodies with knock-out validations. Meanwhile, due to the compensatory effects on Ca$^{2+}$ channels in Ca$_V$1.3$^{-/-}$ and Ca$_V$1.2$^{-/-}$ mice as reported[73,74], cautions also need to be taken in interpreting the data from these knock-out mice. Alternatively, knock-in mice such as Ca$_V$1.2 DHP$^{-/-}$ and/or Ca$_V$1.3 DHP$^{-/-}$ may be advantageous for the purpose of identifying and isolating Ca$_V$1.3 and Ca$_V$1.2 channels.

Our data demonstrate that inhibition of Ca$_V$1/Ca$^{2+}$ influx is highly correlated with attenuation of Ca$_V$1 signaling and neuritogenesis. As mentioned earlier, for particular modulation or perturbation of Ca$_V$1 channels, it may not be as effective as expected for downstream signals and neuritogenesis. In this work, based on the fact that apoCaM/Ca$_V$1 binding is the critical linkage from channel gating to nuclear signaling[20,23,75], we propose DCT/CMI as an outstanding modality with high specificity

and effectiveness compared to other $Ca_V1$ inhibitors known thus far. Particularly targeting $Ca_V1$, DCT consistently generates inhibitory effects across the signaling cascade. We expect that small molecules or biologics mimicking DCT/CMI would provide new interventions for potential therapeutics of diseases related to the $Ca_V1$ activity-neuritogenesis coupling. Both $Ca_V1$ channels and neurite outgrowth are involved in a variety of neuropsychiatric and neurodegenerative diseases, such as autism, bipolar disorder, schizophrenia, Parkinson's disease, and Alzheimer's disease[76–78]. $Ca^{2+}$ dysregulations associated with $Ca_V$ have gained increasing support for its close relevance to neurodegenerative diseases, known as the 'Ca$^{2+}$ hypothesis'[79,80]. However, DCT peptides in these diseases are largely unexplored despite the observations indicating that the amount and distribution of DCT peptides are age-dependent[19,41]. In this regard, there are still unresolved questions pertaining to $Ca_V1$ channels, DCT peptides, and $Ca_V1$/DCT-dependent neuritogenesis. Exemplars of such questions include: whether and why $Ca_V1$ genes (compared with other $Ca_V$) play uniquely important roles in certain pathological processes; whether and how DCT and CMI (compared with other modulations) would play unique roles; whether DCT is prone to disease-associated mutations and how exactly mutant DCT would affect healthy neurons; and eventually what need to do to rescue the defective DCT and neurons. Notably, the expression levels of CaM are downregulated in Parkinson's disease and Alzheimer's disease[77,81], for which the overall inhibitory effects of DCT peptides on $Ca_V1$ and $Ca_V1$-mediated downstream signals would be even more profound due to less apoCaM competition (Fig. 8).

## Methods

**Molecular biology.** The plasmids of channels and peptides were constructed from $\alpha_{1S}$ (rabbit $Ca_V1.1$, NM_001101720.1, Genbank™ accession number), $\alpha_{1C}$ (human $Ca_V1.2$, AF465484.1), $\alpha_{1DL}$ (human $Ca_V1.3$ $\alpha_{1DL}$_human, EU363339.1; or rat $Ca_V1.3$ chimera $\alpha_{1DL}$_rat: backbone a.a. 1-1625 from rat AF370009.1 and $DCT_D$ 1626-2155 from a.a. 1674-2203 of rat NM_017298.1), and $\alpha_{1F}$ (human $Ca_V1.4$ NP005174). In particular, $\alpha_{1\Delta}$ was generated by truncation of $\alpha_{1DL}$ (rat AF370009.1) with a unique XbaI site following the IQ domain (ending with G1625). For chimeric $\alpha_{1\Delta}$-PCRD$_D$ and $\alpha_{1\Delta}$-DCT$_F$, desired segments (PCRD$_D$ from 1626-1780 in $\alpha_{1DL}$_rat and DCT$_F$ from 1596-1966 in $\alpha_{1F}$) were PCR-amplified with SpeI and XbaI sites and cloned into aforementioned $\alpha_{1\Delta}$. Rat $Ca_V1.3$ DHP$^-$ was generated by single point mutation T1033Y[46] on $\alpha_{1\Delta}$, $\alpha_{1DL}$_rat, $\alpha_{1DL}$_V/A (V2075A in $\alpha_{1DL}$_rat) or $\alpha_{1\Delta}$-DCT$_F$, respectively. $\alpha_{1DL}$-Flag was generated by fusing PCR-amplified 3xFlag (DYKDHDGDYKDHDIDYKDDDDK) to the C-terminus of $\alpha_{1DL}$_rat by KpnI and SacII. $\alpha_{1CL}$-Flag was generated by replacing the EGFP in a customized pEGFP-N1 vector (modified by inserting a 3xHA tag before MCS) with PCR-amplified $\alpha_{1CL}$_human fused with 3xFlag via XhoI and NotI sites.

CFP/YFP-DCRD$_F$ in pcDNA3 were constructed as the templates for peptide plasmids. In brief, CFP or YFP was inserted into pcDNA3 vector with the unique KpnI and NotI sites, then DCRD$_F$ was fused to the C-terminus of CFP/YFP by unique NotI and XbaI sites. Other CFP/YFP-tagged constructs were generated by replacing DCRD$_F$ with appropriate PCR amplified segments, via unique NotI and XbaI sites. The constructs we have made include: YFP-DCRD$_F$ truncations, CFP-DCRD$_{S/C/D}$ (DCRD$_D$ from $\alpha_{1DL}$ human EU363339.1), YFP-preIQ$_3$-IQ$_D$-PCRD$_{S/C/D/F}$ (preIQ$_3$-IQ$_D$ from 1576-1625 and PCRD$_D$ from 1626-1780 in $\alpha_{1DL}$_rat), CFP/YFP-PCRD$_{S/C/D/F}$-DCRD$_F$ (PCRD$_D$ from 1626-1780 in $\alpha_{1DL}$_rat), YFP-PCRD$_C$-DCRD$_C$ and YFP-CCAT$_C$. DCRD$_{S/C/D}$-YFP (DCRD$_D$ from $\alpha_{1DL}$_human), CCT$_C$-YFP and CCT$_D$-YFP (CCT$_D$ from $\alpha_{1DL}$_human) were based on another pcDNA3/YFP vector with the cloning sites of KpnI and NotI on 5'. The pcDNA3/YFP vector was made by inserting YFP into pcDNA3 vector with the unique NotI and XbaI sites. Single point mutants such as CFP/YFP-DCRD$_F$_V/A and CCT$_D$_V/A-YFP were made by overlap PCR. For DCT peptides to target nucleus or cytosol, nuclear localization signal (NLS) (PKKKRKV) or nuclear export signal (NES) (LALKLAGLDIGS) was fused to N-terminus of YFP-DCT peptides by overlap PCR, to achieve NLS-YFP-CCAT$_C$, NES-YFP-CCAT$_C$, NLS-YFP-CCT$_C$, NES-YFP-CCT$_C$, NLS-YFP-CCT$_D$ and NES-YFP-CCT$_D$. For 3xFlag-DCT$_{C/D}$, 3xFlag tag was firstly inserted into pcDNA3 vector with the unique KpnI and NotI sites, then DCT$_{C/D}$ (from $\alpha_{1CL}$_human or $\alpha_{1DL}$_human) peptides were PCR-amplified with NotI and XbaI sites and fused to the C-terminus of 3xFlag.

**Dissection and culturing of cortical neurons.** Cortical neurons were dissected from postnatal day 0 (P0, either sex) newborn ICR mice. Isolated cortex tissues were digested with 0.25% trypsin for 15 min at 37 °C, followed by terminating the

enzymatic reaction by DMEM supplemented with 10% FBS. The suspension of cells was sieved through a filter then centrifuged at 1000 rpm for 5 minutes. The cell pellet was resuspended in DMEM supplemented with 10% FBS and then plated on poly-D-lysine-coated 35-mm No. 0 confocal dishes (In Vitro Scientific) or poly-D-lysine-coated coverslips. After 4 hours, neurons were maintained in Neurobasal medium supplemented with 2% B27, 1% glutaMAX-I (growth medium). Temperature of 37 °C with 5% $CO_2$ was controlled in the incubator. All animals were obtained from the laboratory animal research centers at Tsinghua University and Peking University. Procedures involving animals have been approved by local institutional ethical committees of Tsinghua University and Beihang University.

**Transfection of cDNA constructs in cell lines and cultured neurons.** For electrophysiological recording, HEK293 cells (ATCC), checked by PCR with primers 5′-GGCGAATGGGTGAGTAACACG-3′ and 5′-CGGATAACGCTTGCGA CCTATG-3′ to ensure free of mycoplasma contamination, were cultured in 60-mm dishes, and recombinant channels were transiently transfected according to established calcium phosphate protocol[20,24]. 5 μg of cDNA encoding channel $\alpha_1$ subunit, along with 4 μg of rat brain $\beta_{2a}$ (M80545) and 4 μg of rat brain $\alpha_2\delta$ (NM012919.2) subunits were applied to HEK293 cells. To enhance expression, cDNA for simian virus 40 T antigen (1 μg) was also co-transfected. For each additional construct, 2 μg cDNA was added. All of the above cDNA constructs were driven by a cytomegalovirus (CMV) promoter. Cells were washed with PBS 6–8 h after transfection and maintained in supplemented DMEM, then incubated for at least 48 h in a water-saturated 5% $CO_2$ incubator at 37 °C before usage.

For transfection in neurons, 2 μg of cDNA encoding the desired peptides were transiently transfected by Lipofectamine 2000 (Invitrogen) for each confocal dish with a typical protocol according to the manual. The mixture of plasmids and Lipofectamine 2000 in opti-MEM was added to the Neurobasal medium for transfection. After 2 hours, neurons were maintained in Neurobasal medium supplemented with 2% B27, 1% glutaMAX-I for 48 hours.

For 2-hybrid 3-cube FRET experiments, HEK293 cells were cultured on confocal dishes. FRET cDNA constructs of 2 μg each were transfected by Lipofectamine 2000 for 6 hours. Cells were used after 24 hours.

For western blot experiments, CHO (Cell Resource Center, IBMS, CAMS/PUMC) or HEK293 cells were cultured on 60 mm dishes. cDNA constructs were transfected by Lipofectamine for at least 6 hours. Cells were collected after 2 days.

**Whole-cell electrophysiology.** Whole-cell recordings of transfected HEK293 cells were performed at room temperature (25 °C) using an Axopatch 200B amplifier (Molecular Devices). Electrodes were pulled with borosilicate glass capillaries by a programmable puller (P-1000, Sutter Instrument) and heat-polished by a microforge (MF-830, Narishige), resulting in 2–5 MΩ resistances before 70% of compensation. The internal/pipette solution contained (in mM): CsMeSO$_3$, 135; CsCl, 5; MgCl$_2$, 1; MgATP, 4; HEPES, 5; and EGTA, 5; with ~290 mOsm adjusted with glucose and pH 7.3 adjusted with CsOH. The extracellular/bath solution contained (in mM): TEA-MeSO$_3$, 135; HEPES, 10; CaCl$_2$ or BaCl$_2$, 10; with ~300 mOsm, adjusted with glucose and pH 7.3 adjusted with TEA-OH, similar to the previous protocols[24]. Whole-cell currents were generated from a family of step depolarizations (−70 to +50 mV from a holding potential of −70 mV and step increase of 10 mV). Current traces were recorded at 2 kHz low-pass filtering in response to voltage steps with minimum interval of 30 s. P/8 leak subtraction was used throughout. $Ca^{2+}$ current was normalized over different cells by cell capacitance ($C_m$, in pF), and the current amplitude (peak, 50 ms or 300 ms, in pA/pF) was measured at −10mV.

Neuronal patch-clamp recording was performed according to our previous protocol[51]. In brief, isolated cortical neurons were cultured in coverslips. To record neuronal $Ca_V1.3$ current, neurons were pre-incubated in Tyrode's solution containing 1 μM nimodipine (Sigma-Aldrich), 1 μM ω-conotoxin GVIA (Sigma-Aldrich, or alomone labs) and 1 μM ω-conotoxin MVIIC (Sigma-Aldrich, or alomone labs) for 30 min to block endogenous $Ca_V1.2$, N- and P/Q-type $Ca^{2+}$ current, according to the cocktail recipes[75,82,83]. Under the conditions of our evaluations (−10 mV, full cocktail recipes), $Ca_V1.3$ current appeared to be the dominant component (~80%) after the treatment, and $Ca_V2.3$ (16%) and $Ca_V1.2$ (4%) contributed to the rest (details see Supplementary Fig. 2). The voltage ramp protocol (holding at −60 mV, ramping from −60 to +50 mV at 0.2 mV/ms) was applied to cortical neurons in the bath solution containing 10 mM $Ba^{2+}$. The resulted I-V curves were fitted by Boltzmann-based equations (OriginPro) to obtain the half-activation voltage ($V_{half}$) of voltage-dependent channel activation. Isradipine (Sigma-Aldrich) at 100 nM was used to further isolate/confirm $Ca_V1.3$ currents, and 20 μM isradipine would eliminate all the $Ca_V1$ ($Ca_V1.2$ and $Ca_V1.3$) currents[84]. Treated neurons were recorded in various bath solutions containing appropriate blockers within one hour.

**2-hybrid 3-cube FRET.** 2-hybrid 3-cube FRET experiments were carried out with standard protocols similarly shared by several groups[20,24,56]. Briefly, experiments were performed on an inverted epi-fluorescence microscope (Ti-U, Nikon), with computer-controlled filter wheels (Sutter Instrument) to coordinate with diachronic mirrors for appropriate imaging at excitation, emission, and FRET channels. The filters used in the experiments were excitation: 438/24 (FF01-438/24-25,

Semrock) and 480/30 (FITC, Nikon); emission: 483/32 (FF01-483/32-25, Semrock) and 535/40 (FITC, Nikon); dichroic mirrors: 458 nm (FF458-Di02-25 × 36, Semrock) and 505 nm (FITC, Nikon). Fluorescence images were acquired by Neo sCMOS camera (Andor Technology) and analyzed with $3^3$-FRET algorithms coded in Matlab (Mathworks), mainly based on the following formula:

$$FR = 1 + \frac{FR_{max} - 1}{1 + \frac{K_d}{D_{free}}} \qquad (E1)$$

$FR_{max}$ represents the maximum FRET ratio, and $D_{free}$ denotes the equivalent free donor (CFP-tagged) concentration. $K_d$ (effective dissociation equilibrium constant) is calculated from an iterative procedure to evaluate the binding affinity for each pair of binding partners. FRET imaging experiments were performed with HEK293 cells in Tyrode's buffer containing 2 mM $Ca^{2+}$.

**Confocal fluorescence imaging and analysis.** Cultured neurons were transfected with CFP (to label the soma area and neurites) and DCT peptides tagged with YFP on 5th day (DIV-5) and used on DIV-7, or transfected on DIV 12-15 and used on DIV 15–18 (Supplementary Fig. 10). Neurons were loaded with Hoechst 33342 for 5 min to label the nuclei and then imaged by Zeiss LSM710 confocal Scanning Microscope. Fluorescent intensity was quantified and analyzed with ImageJ (NIH). Calculation of nuclear intensity was based on the nuclear contour indicated by Hoechst 33342. Cytosolic intensity was calculated by intermediate region between nucleus and plasma membrane. N/C ratio of DCT peptides was calculated by the ratio of fluorescence intensity (nuclear/cytosolic). Measurements of the total length and *Sholl* analysis for neurites were performed with Imaris 7.7.2 (Bitplane) through CFP channel. Only non-overlapping neurons were selected for analysis of morphogenesis. Neurite tracings were depicted with Imaris 7.7.2 and further processed with Photoshop 7.0 (Adobe).

To observe the cytosolic-nuclear translocation of DCT peptides, neurons were pre-incubated in 5 mM $[K^+]_o$ solution (130 mM NaCl, 5 mM KCl, 1 mM $MgCl_2$, 15 mM HEPES, 2 mM $CaCl_2$, at 300 mOsm adjusted with glucose) and perfused with 40 mM $[K^+]_o$ solution (95 mM NaCl, 40 mM KCl, 1 mM $MgCl_2$, 15 mM HEPES, 2 mM $CaCl_2$, at 300 mOsm adjusted with glucose) or 5 mM $[K^+]_o$ with 50 μM Nifedipine for 0.5-1 hour, then washed out by 5 mM $[K^+]_o$ when needed. For the experiments with DHP-insensitive variants of $Ca_V1$, neurons were incubated with 50 μM Nifedipine for at least 1 hour, and neurons without clear damages were selected to calculate N/C ratio for the peptides.

Analyses on neurite morphology and cytosolic-nuclear translocation were performed over cultured neurons from at least two culture preparations and two independent experiments, adding up to the total number for each data group (20 cells or more).

**Immunocytochemistry.** DIV-5 cultured cortical neurons were transfected with $DCRD_F$ and used on DIV-7. Firstly, to stimulate neurons, 1 μM TTX (sodium channel blocker) was applied to neurons for 6 hours to suppress action potential. 40 mM $[K^+]_o$ solution containing 1 μM TTX was applied for 5 min for CaM staining, or 30 min for pCREB staining before fixation. To measure desired signals in physiological condition, neurons were maintained in growth medium until fixation. Secondly, neurons were rapidly rinsed with ice-cold PBS and fixed with ice-cold 4% paraformaldehyde in PBS (pH 7.4) for 15–20 min. Fixed neurons were washed by ice-cold PBS for 3 times and permeabilized with 0.3% Trition X-100 for 5 minutes. Then neurons were blocked by 10% normal goat serum in PBS for 1 hour and incubated with the primary antibodies overnight at 4 ºC. The following antibodies were used: CaM (Rabbit mAb #5197-1, Epitomics, Species Cross-Reactivity: Human, Mouse, Rat, Dilutions: 1:500 in PBS)[51], pCREB (Rabbit mAb #9198, Cell Signaling Technology, Species Cross-Reactivity: Human, Mouse, Rat, Dilutions: 1:500 in PBS)[49,51], c-Fos (Rabbit mAb [EPR21930-238] #ab222699, Abcam, Species Cross-Reactivity: Mouse, Human, Dilutions: 1:1000 in PBS, https://www.abcam.com/nav/primary-antibodies/rabbit-monoclonal-antibodies/c-fos-antibody-epr21930-238-ab222699.html), and $Ca_V1.3$ CT (a.a. 2025-2161) (Mouse mAb [N38/8] #ab84811, Abcam, Species Cross-Reactivity: Mouse, Rat, Rabbit, Human, Dilution: 1:500 in PBS)[85]. Finally, the next day, neurons were washed with PBS for 3 times and incubated with the secondary antibodies (Goat anti-Rabbit Alexa Fluor 647, Invitrogen, Dilutions: 1:800 in PBS; Goat anti-Mouse Alexa Fluor 568, Invitrogen, Dilutions: 1:800 in PBS; Goat anti-Mouse IgG(H+L)-DyLight 488, Gene-Protein Link, Dilutions: 1:500) for 2 hours. Then neurons were washed with PBS for 3 times and treated with Hoechst 33342 (Invitrogen) for 5 min for nuclear counterstain. Mounted neurons on confocal dishes were imaged with a confocal microscope (LSM710, Carl Zeiss) and ZEN software. Nuclear and cytosolic fluorescence intensities of endogenous CaM in cortical neurons were analyzed by ImageJ (NIH). Neuronal culture preparations of each round supplied one or two independent experiments. Neurons (>15) were assayed or evaluated from at least two culture preparations and two independent experiments unless otherwise noted.

**Western blot.** Cortical neurons for western blot were isolated from newborn ICR mice or from cultured cortical neurons around DIV-14. Liver tissues were from newborn ICR mice. HEK293 Cells and CHO (Chinese Hamster Ovary) cells served as the blank control or the recombinant system to express proteins. Cells lysates were prepared by lysis buffer RIPA (with protease inhibitor cocktail, cat#P1265,

Applygen Tech) and centrifuged at 15,000 × g. Proteins were separated using 8% sodium dodecyl sulphate polyacrylamide gel electrophoresis and transferred to a nitrocellulose membrane for 100 min. Then nitrocellulose membrane was blocked in 5% non-fat dry milk and incubated with primary antibodies: anti-$Ca_V1.3$ CT, identical to the antibody used in immunocytochemistry, Dilutions: 1:1000; anti-$Ca_V1.2$ CT (a.a. 1835-2135) (Rabbit pAb #21774-1-AP, proteintech, Species Cross-Reactivity: human, mouse, rat, Dilutions: 1:1000, https://www.ptgcn.com/products/L-VOCC-Antibody-21774-1-AP.htm); anti-$Ca_V1.2$ II-III loop (#ACC-003, allomone, Source: Rabbit, Species Cross-Reactivity: Human, Mouse, Rat, Dilutions: 1:1000, https://www.alomone.com/p/anti-cav1-2-antibody/ACC-003), anti-Flag (#20543-1-AP, proteintech, Source: Rabbit, Species Cross-Reactivity: recombinant protein with Flag tag, Dilutions: 1:3000); and anti-GAPDH (#P01L081, Gene-Protein Link, Source: Rabbit, Species Cross-Reactivity: Human, Mouse, Rat, Bovine, Pig, Chicken, Zebrafish, Green Monkey, Dilutions: 1:5000) overnight at 4 °C. Next, the nitrocellulose membrane was washed three times with TBST and incubated with secondary antibody (Goat anti-Mouse, # SA00001-1, Proteintech, Dilution: 1:3000; Goat anti-Rabbit, #ZRA03, YTHXbio, Dilution: 1:5000) for 1–2 h and then washed with TBST for three times again. The membrane was coved with ECL chemiluminescent liquid (#P1010, Applygen Tech) before detection with an enhanced chemiluminescence system. Three or more independent replicates were performed for each experiment.

**Definition of CMI potency and related curve fitting.** For $α_{1DL}$ channels with covalently-linked DCT (Figs. 2, 7), there are two subgroups: DCT-bound and apoCaM-bound. For these channels subject to modulation by standalone DCT peptides, CMI potency (*CMI*, in percentage) is quantified as the normalized fractional change of apoCaM-bound channels switching to peptide-bound channels ($f_{Peptide}$):

$$CMI = f_{Peptide} = \frac{\Delta F_{CaM}}{F_{CaM}} \qquad (E2)$$

Since apo-CaM bound fraction ($F_{CaM}$) is proportional to $Ca^{2+}$-dependent inactivation ($S_{Ca}$)[20], we have:

$$CMI = \frac{S_{Ca,Control} - S_{Ca,Peptide}}{S_{Ca,Control}} = 1 - \frac{1}{S_{Ca,Control}} \cdot S_{Ca,Peptide} \qquad (E3)$$

Hence, CMI potency is inversely and linearly correlated with actual inactivation of channels under peptide modulation. Meanwhile, interactions between peptides and channels should follow the relationship defined by the classical ligand-binding equation:

$$CMI = f_{Peptide} = \frac{[Peptide]}{[Peptide] + K_d} \qquad (E4)$$

Here, [Peptide] is the concentration of DCT peptide, and $K_d$ is the dissociation constant of peptide interactions with channels. Fluorescence intensities (a.u., arbitrary units) are used to calculate [Peptide] and $K_d$. In our electrophysiology experiments, patch-clamped cells are supposed to have comparable (over) expression levels of peptides, thus Eq. E4 represents the relationship between *CMI* and $K_d$ across a series of peptide variants (Fig. 4c). The concentration of DCT peptides ([Peptide]) on average was estimated to be ~23,000 (a.u.), by directly applying Eq. E4 to fit Fig. 4c. To estimate the binding affinities between YFP-preIQ3-IQD and CFP-PCRDX-DCRDF, we proceeded to perform FRET experiments between YFP-preIQ3-IQD-PCRDX and CFP-DCRDF. The reason for taking such indirect approach is in part due to severe aggregation (puncta) for YFP-preIQ3-IQD (but not for YFP-preIQ3-IQD-PX) when overexpressed in cells. Correspondingly, *CMI* values of PCRDX-DCRDF peptides to $α_{1DA}$ were from Fig. 4e. $K_d$ for IQD and PD-DF was estimated to be 25-fold of the $K_d$ value listed for IQD-PD and DF in Fig. 4c. Similar calculations (25-fold) can be conducted to estimate $K_d$ values for other peptides in Fig. 4f from Supplementary Fig. 7.

In Fig. 4g, results from Fig. 4c, f were combined and plotted together with the tuning curve (Eq. E4, ([Peptide] = 23,000). In addition, compound CMI effects for PD+PF-DF, PD+PC-DF, and PD+PC-DC were also illustrated, based on *CMI* values from electrophysiology and estimated $K_d$ values by interpolation.

Regarding full-length channels with covalently-linked DCT of its own (such as $α_{1DL}$ and DCT mutants), DCT potency (of competing with apoCaM to bind the channel) can be derived from Eqs. E2 and E3 into similar formulas for the intramolecular CMI by the covalently-linked DCT (Figs. 5j, 7d). Considering $α_{1DΔ}$ as the control (100% apoCaM-bound), (CMI) potency of linked DCT domains represents the fraction of channels bound with DCT ($f_{DCT}$) for particular channel/DCT variant. We have

$$CMI = f_{DCT} = 1 - F_{CaM} = 1 - \frac{S_{Ca,DCT}}{S_{Ca,Control}} \qquad (E5)$$

In principle, *CMI* refers to fractional changes of channels switched from apoCaM-bound to apoCaM-unbound (i.e., DCT-bound), which could be achieved by standalone DCT peptides and/or covalently-linked DCT domains in a similar fashion.

**Statistics and reproducibility.** Data were analyzed in Matlab, OriginPro and GraphPad Prism software. Data were shown as mean ± SEM (Standard Error of the

Mean). Unpaired or paired Student's $t$-test (two-tailed with criteria of significance) was performed to compare two groups. One-way ANOVA followed by Dunnett or Bonferroni for post hoc tests were performed to compare more than two groups with or without a restrictive control group, respectively, provided the normal distribution of the data. Kruskal-Wallis and Dunn's non-parametric test was performed if the data did not follow the normal distribution. D'Agostino & Pearson omnibus normality test was used before column analyses. Significance $*p < 0.05$; $**p < 0.01$; $***p < 0.001$ and $n.s.$ denotes 'not significant'.

**Reporting summary**. Further information on research design is available in the Nature Research Reporting Summary linked to this article.

## Data availability

The plasmids of pcDNA3-NLS-YFP-CCT$_D$ (#184325), pcDNA3-NES-YFP-CCT$_D$ (#184326), pcDNA3-NLS-YFP-CCT$_C$ (#184327) and pcDNA3-NES-YFP-CCT$_C$ (#184328) are available on Addgene. Source data underlying the figures are organized as Supplementary Data 1. Uncropped blots are presented in Supplementary Fig. 17. The data in details associated with the main figures have been deposited to Dryad (https://doi.org/10.5061/dryad.cvdncjt63)[86]. Other data and information are available from the corresponding author upon reasonable request.

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

## Acknowledgements
We thank all X-Lab members for discussions and help. This work is supported by grants from Natural Science Foundation of China (81971728 and 21778034 for XDL, 11902021 for YXY, U20A20390 and 11827803 for YBF) and Natural Science Foundation of Beijing Municipality (7191006 for XDL, 5204037 for YXY) and China Postdoctoral Science Foundation (BX20180027 and 2018M641146 for YXY), and by an open fund from Laboratory for Biomedical Engineering of Ministry of Education, Zhejiang University.

## Author contributions
X.D.L. and Y.X.Y. conceived the project; X.D.L., Y.B.F., X.M.L., S.S., and P.W. provided general help; M.L. and N.L. made important contributions by conducting the pilot experiments and analyses; Y.X.Y., Z.Y., and J.L.G. performed the experiments; Y.X.Y. and X.D.L analyzed the data; P.L., W.L.H., S.H.Y., H.J., H.Y.G., F.Q., and W.X. provided technical assistance; X.D.L. and Y.X.Y. wrote the paper.

## Competing interests
The authors declare no competing interests.
