## [Peer Review File · Communications Biology]

Reviewers' comments:

Reviewer #1 (Remarks to the Author):

The manuscript by Yang et al. made a thorough investigation on how the distal C-terminus (DCT) of Cav1 influenced Cav1 channel activity and downstream signaling. By comparing the effects of DCT domains with different domains, they found that DCRDF and DCRDD could potentially inhibit Cav1 channel activity, CREB phosphorylation, c-fos expression and neurite outgrowth. Mechanistically, they found that these effects were due to CMI (C-terminus mediated inhibition). They further found that DCT peptides expressed in the cytosol or nucleus had opposite effects on neurite outgrowth. Both endogenous and exogenous cytosolic DCT peptides induced neurite retraction. The manuscript was well-written. The data clearly and convincingly support the authors' hypothesis that the cytosolic DCT peptides downregulate excitation-neuritogenesis coupling. Their findings solved the long-term puzzle of why different groups found very different effects of DCT peptides on Cav channel activity and calcium signaling. My only suggestion to the authors is that they had better confirm that DCRDF or DCRDD has no acute effect on Cav channel membrane expression.

Reviewer #2 (Remarks to the Author):

This manuscript demonstrates that the standalone CaV1-DCTs bind to CaV1 channels, inhibit Ca influx presumably by competing with CaM, and inhibit neurite development. The work is based on cellular imaging and electrophysiology with a broad array of different conditions and molecular constructs to identify the motifs in CaV1-DCTs that are important for binding to the channel, the role of CaV1-DCTs in inhibiting vs the previous known promoting neurite development depending on their cellular localization (cytosolic vs nuclear), and how Ca²⁺ influx regulates CaV1-DCTs cellular localization. This is a comprehensive study of the mechanism of CaV1-DCTs in regulating excitation-neuritogenesis, with a thorough approach and high quality data. The results will be interesting to a broad audience. While the authors should be commended for presenting such a complex study with clear logic so that it can be understood, the complex experimental conditions and various constructs can make it a struggle for readers, particularly for those who are not expert in Ca channel field. Something need to be done to make it easier to access. For instance, some data may be moved to supplemental figures to lower the barrier for the readers to get to the essence of the figures. Alternatively, a cartoon representing the essence of the figure can be presented in some of the figures.

Reviewer #3 (Remarks to the Author):

Review of "Cytosolic peptides encoding C-termini of CaV1 channels downregulate excitation-neuritogenesis coupling" by Yaxiong Yang et al.

Brief Summary:

In this manuscript, the authors investigate the molecular mechanisms whereby the distal C-terminal domain (DCT) of the Cav1 L-type channel regulates the functional properties of the channel in neurons and the roles of this channel in neuritogenesis. They test the hypothesis that the DCT-dependent inhibition of Cav1 channels suppresses Cav1 mediated neuritogenesis. There are two main mechanisms that are explored, one is that the Cav1 DCT reduces Ca²⁺ influx by competing with calmodulin (CaM) binding to the proximal C-terminal site thus inhibiting both inactivation and activation. The second mechanism is the stand-alone function of the Cav1 DCT as an independent peptide (cleaved or independently transcribed-- calcium channel associated transcription regulator (CCAT)) with effects on transcription. Additionally, the authors test how Ca²⁺ influx through Cav1 channels regulates the subcellular DCT distributions in neurons. While the structure/function analysis of the DCT in relation to channel biophysics is well-executed, the tie-in of these results with those of the neuronal experiments is largely correlational, and the latter experiments lack critical controls. In addition, the authors seem to conflate activity-dependent effects of Cav1 channels with those due to basal neuronal activity, and do not reconcile their

findings with previously described actions of Cav1 blockade on promoting rather than inhibiting neurite growth. Finally, description of methodology and statistical analyses lack sufficient detail to evaluate the rigor of the approach.

Major comments:

- 1) Based on parallel studies of the effects of DCT fragments on Cav1.3 function and neuritogenesis, the authors conclude that the DCT fragments inhibit neuritogenesis through actions on Cav1.3 activation and inactivation. This is problematic first because the possibility that the DCT fragments regulate Cav1.2-dependent signaling cannot be discounted. The pharmacological dissection of Cav1.3 is not convincing: in Supp Fig.2, the authors show that a "cocktail" of inhibitors including 10 μ M nimodipine incompletely blocks the whole cell I_{Ca}. Because the L-type blocker isradipine (20 μ M) but not the R-type blocker SNX-482 blunts the residual current, the authors conclude this approach effectively isolates the Cav1.3 contribution. The data are derived from drug effects on I_{Ca} evoked by only one test pulse. Because dihydropyridine block is voltage-dependent, analysis at a single pulse could underestimate the contribution of Cav1.2 to the whole cell I_{Ca} in the cortical neurons in Supp fig2A, such that Cav1.2+Cav1.3 mediate the isradipine-sensitive current. Cav1.3 KO or DHP-insensitive Cav1.2 mutant mice are available and could readily address this caveat (there is also the wrinkle caused by splice variation of Cav1.2 and Cav1.3 affecting dihydropyridine sensitivity in neurons). Fig. 2A shows the effects of the DCRDF on I_{Ca} evoked by -10 mV, and the authors conclude that the reduction in the current amp is consistent with the effects of DCRDF on inhibiting activation. The authors cannot discount that DCRDF affects channel trafficking to the membrane-- why not show the IV o back up their interpretation? This is a concern since G. Pitt's group showed that blunting CaM interactions with Cav1 can inhibit cell-surface and dendritic trafficking these channels in neurons.
- 2) Despite the focus on Cav1.3, the neuronal experiments utilize strategies that would not distinguish between Cav1.2 and Cav1.3. In Figs2C,D, high K⁺ (which will activate both Cav1.2 and Cav1.3) is used to analyze nuclear CaM and pCREB. How this signaling pathway might contribute to neuritogenesis analyzed in Fig.2B and c-fos (fig.2E) is not clear (these assays did not utilize high K).
- 3) There is rather loose citing of past work in support of the authors' interpretations. Redmond, Ghosh show that Cav1-mediated excitation-transcription coupling regulates activity-dependent neurite growth (i.e., that promoted by high K⁺). However, the paradigm here looks at effects of DCT fragments on basal neurite growth. Krey, Dolmetsch show that a Timothy syndrome mutation that suppresses inactivation causes dendrite retraction as opposed to the stimulatory effects of DCT peptides on neurite growth studied here.
- 4) Relevant to the previous point, pharmacological and genetic suppression of Cav1 activity is known to promote rather than inhibit neurite regeneration in some contexts (PMID: 23313911, 20579880, 30232777, 28320840). The authors should reconcile their findings with this past work.
- 5) Results in Fig.6 investigating the nuclear localization dynamics of the endogenous CTD peptide are compromised by the lack of rigor in characterizing the specificity of the CTD antibodies. Western blot in Fig.6A showing band detected in mouse brain but not untransfected CHO cells does not discount possibility that what is detected by IF in B-D is unrelated to Cav1.3 (the paper cited in the methods for characterizing the Cav1.3 antibody used did not use appropriate negative controls for specificity). The authors imply that the 55-kDa fragment results from proteolysis of the channel, but this has not been shown definitively for mouse cortical neurons.
- 6) Statistical analysis is improperly documented and sometimes absent (i.e., supp Fig.2, all Sholl analyses). Tests to verify normality of the data are warranted for the tests used (t-tests, ANOVA).

Minor comments:

- 1) Include the number of independent cultures per experiment along with the number of cells from each cortical neuron culture. The cell numbers are different in all experiments, so it is hard to follow the cell numbers from each independent culture.
- 2) Sholl analysis is often compared as the area under the curve (AUC) which would help back up claims re: significant differences in Fig. 1b, Fig. 3c and Fig. 5 c and g
- 3) Text could use editing for grammar.
- 4) Double check text matches the figure references. Page 13, line 362 has a mislabeled figure number it reads (Fig. 6i, j) which I think refers to (Fig. 5i, j). Same mistake in page 15, line 414. Page 20, line 556 refers to supplementary Fig. 9 for the distinctions between the short and

long/medium peptides demonstrate but that figure only shows the N/C ratio of DCRDC/D. I think Fig 5 h and I are the proper figure reference.

Reviewers' comments:

Reviewer #1 (Remarks to the Author):

The manuscript by Yang et al. made a thorough investigation on how the distal C-terminus (DCT) of Cav1 influenced Cav1 channel activity and downstream signaling. By comparing the effects of DCT domains with different domains, they found that DCRDF and DCRDD could potentially inhibit Cav1 channel activity, CREB phosphorylation, c-fos expression and neurite outgrowth. Mechanistically, they found that these effects were due to CMI (C-terminus mediated inhibition). They further found that DCT peptides expressed in the cytosol or nucleus had opposite effects on neurite outgrowth. Both endogenous and exogenous cytosolic DCT peptides induced neurite retraction. The manuscript was well-written. The data clearly and convincingly support the authors' hypothesis that the cytosolic DCT peptides downregulate excitation-neuritogenesis coupling. Their findings solved the long-term puzzle of why different groups found very different effects of DCT peptides on Cav channel activity and calcium signaling. My only suggestion to the authors is that they had better confirm that DCRDF or DCRDD has no acute effect on Cav channel membrane expression.

We are very grateful to the reviewer for the positive comments and the constructive suggestion. Our previous study has demonstrated that the C-terminus mediated inhibition (CMI) by DCT peptide acts on channel gating in an acute manner, which is quantitatively indistinguishable from the 'chronic effects' by overexpressing DCT fragments (Liu, 2017 eLife). Here, as suggested by the reviewer, we have provided direct evidence that DCT peptides do not alter channel expression or membrane trafficking (**Supplementary Fig. 14c**). When cortical neurons were overexpressed with the DCT fragments from Cav1.3 in cultured cortical neurons, neither the total expression nor the (membrane) surface expression of Cav1.3 exhibited any change in our immunostaining experiments (using the antibody targeting the extracellular loop of the channel). Altogether, we have concluded that DCT peptides through their direct binding to Cav1.3 produce no or negligible effect on the expression of Cav1.3.

Reviewer #2 (Remarks to the Author):

This manuscript demonstrates that the standalone CaV1-DCTs bind to CaV1 channels, inhibit Ca influx presumably by competing with CaM, and inhibit neurite development. The work is based on cellular imaging and electrophysiology with a broad array of different conditions and molecular constructs to identify the motifs in CaV1-DCTs that are important for binding to the channel, the role of CaV1-DCTs in inhibiting vs the previous known promoting neurite development depending on their cellular localization (cytosolic vs nuclear), and how Ca²⁺ influx regulates CaV1-DCTs cellular localization. This is a comprehensive study of the

mechanism of Cav1-DCTs in regulating excitation-neuritogenesis, with a thorough approach and high quality data. The results will be interesting to a broad audience. While the authors should be commended for presenting such a complex study with clear logic so that it can be understood, the complex experimental conditions and various constructs can make it a struggle for readers, particularly for those who are not expert in Ca channel field. Something need to be done to make it easier to access. For instance, some data may be moved to supplemental figures to lower the barrier for the readers to get to the essence of the figures. Alternatively, a cartoon representing the essence of the figure can be presented in some of the figures.

We very much appreciate the reviewer for the encouraging comments on our work. Following the suggestions, a portion of DCRD_F data are now moved to **Supplementary Fig. 3** and **Supplementary Fig. 4**. DCRD_F effects on Cav1 signaling under high K⁺ are now removed from the main figure (**Fig. 1**) for clarity, and the related evaluations of CMI and S_{Ca} are removed from **Fig. 2**, since DCRD_F is not necessarily as quantitative as the latter five DCT peptides in **Fig. 2a**. Additional cartoons are placed side by side for each type of DCT peptides in **Fig. 2b**, where the potential differences are illustrated. In addition, in **Fig. 6**, the data of exogenous DCT peptides are moved to **Supplementary Fig. 14a, b**, so that the main figure is devoted to the endogenous Cav1.3-encoded peptides newly discovered by this work. Moreover, we have gone through the manuscript to adjust our wording/presentations in order to lower the reading barriers.

Reviewer #3 (Remarks to the Author):

Review of "Cytosolic peptides encoding C-termini of Cav1 channels downregulate excitation-neuritogenesis coupling" by Yaxiong Yang et al.

Brief Summary:

In this manuscript, the authors investigate the molecular mechanisms whereby the distal C-terminal domain (DCT) of the Cav1 L-type channel regulates the functional properties of the channel in neurons and the roles of this channel in neuritogenesis. They test the hypothesis that the DCT-dependent inhibition of Cav1 channels suppresses Cav1 mediated neuritogenesis. There are two main mechanisms that are explored, one is that the Cav1 DCT reduces Ca²⁺ influx by competing with calmodulin (CaM) binding to the proximal C-terminal site thus inhibiting both inactivation and activation. The second mechanism is the stand-alone function of the Cav1 DCT as an independent peptide (cleaved or independently transcribed--calcium channel associated transcription regulator (CCAT)) with effects on transcription. Additionally, the authors test how Ca²⁺ influx through Cav1 channels regulates the subcellular DCT distributions in neurons. While the structure/function analysis of the DCT in

relation to channel biophysics is well-executed, the tie-in of these results with those of the neuronal experiments is largely correlational, and the latter experiments lack critical controls. In addition, the authors seem to conflate activity-dependent effects of Cav1 channels with those due to basal neuronal activity, and do not reconcile their findings with previously described actions of Cav1 blockade on promoting rather than inhibiting neurite growth. Finally, description of methodology and statistical analyses lack sufficient detail to evaluate the rigor of the approach.

We thank the reviewer for these detailed comments and insightful opinions. Some of the items will be individually addressed in responses to the “Major” and “Minor” comments.

Before further elaborations, we would like to introduce the following key points:

1. **Inter-** versus **Intra-molecular** DCT effects.

We have noticed that the reviewer was confused with the intra- versus inter-molecular DCT effects (see above “two main mechanisms”). *Intra-molecular* DCT effects have been reported by earlier studies on the role of DCT motif within the channel. Here, we focus on the effects of standalone DCT peptides especially when these peptides are present in the cytosol (versus in the nucleus), for which the *inter-molecular* binding between peptides and the channel is essential. Nevertheless, the two kinds of effects share the same mechanisms of action: to inhibit channel gating by competing against apoCaM at the preIQ-IQ domain. DCT peptides in the nucleus promote the expression of essential genes directly as transcription factors, a distinct scenario compared with cytosolic DCT peptides that cause inhibitory effects on Cav1/Ca²⁺-dependent neuritogenesis.

2. Neuritogenesis for cortical neurons under **Basal** versus **Excited** conditions.

We have followed the convention that “activities” normally refer to additional stimuli/depolarization on top of basal conditions, which would indeed lead to pronounced dendrite outgrowth (Redmond 2002 Neuron, PMID: 12086646). Meanwhile, it is also documented that spontaneous Ca²⁺ waveforms are observed from cultured neurons where voltage-gated calcium channels are closely involved (Uhlen 2010 BBRC, PMID: 20494106). These “basal” activities of Cav1 channels particularly in the form of autonomous Ca²⁺ oscillations at relatively low frequency play important roles in neuronal development including neuritogenesis (Tang 2003 JN, PMID: 12574421; Gomez 2006 Nat Rev Neuroscience, PMID: 16429121; Toth 2016 Cell Calcium, PMID: 27020657; Kamijo 2018 JNS, PMID: 29773754). Thus, in the context of our work, Ca²⁺ signals at the basal conditions are dominantly mediated by Cav1 channels in neurons, especially when membrane potentials fall into the range of -20 mV and below (Wheeler 2012 Cell, PMID: 22632974). We agree that the channel activities could be further enhanced by physiological stimuli as mimicked by high K⁺, leading to neurite extension and arborization, conventionally described as activity-dependent neuritogenesis (Redmond 2002 Neuron, PMID: 12086646). In principle, there is no fundamental difference between the channels

working in “excited” versus “basal” states, since in either state channels are activated by the voltage rise to allow Ca^{2+} influx but just of different amount/strength. Through CMI, the DCT peptides inhibit Cav1 of either state (basal or stimulated) by the same mechanisms. On some occasions, we conducted experiments under both 5 mM K^+ (**Fig. 1**) and 40 mM K^+ (**Supplementary Fig. 3**) conditions, both leading us to the same conclusions. And as the reviewer pointed out, we mostly rely on “basal” Cav1 activities (spontaneous and oscillatory Ca^{2+} influx via Cav1 channels) which are physiologically linked to neuritogenesis of cortical neurons.

3. **Coupling and Decoupling** between Cav1 activities and downstream signaling.

Complicated and even contradictory results have been reported on the cellular effects of pharmacological interventions of Cav1. In fact, one major advantage of CMI/DCT peptides has been proposed in Introduction. That is, Cav1 inhibitors just simply targeting the Ca^{2+} influx may not be able to modulate Cav1-dependent neuritogenesis as expected (decoupling between Ca^{2+} influx and Cav1 signaling). DCT peptides of Cav1 compete with CaM at IQ, both critical to Cav1 downstreams (e.g. CREB), as the potential assurance for DCT peptides to effectively inhibit neuronal signaling. On the contrary, the inhibition by blockers (e.g., Cd^{2+} versus nimodipine) or facilitation by TS mutations (G406R versus G402S) solely defined by channel gating may not ensure the down- or up-regulations of Cav1 signaling (Li 2016 Science, PMID: 26912895; Krey 2013 Nature Neuroscience, PMID: 23313911). It needs case-by-case investigations to elucidate the mechanism of decoupling for Cav1 modulators or defects.

Major comments:

1) *Based on parallel studies of the effects of DCT fragments on Cav1.3 function and neuritogenesis, the authors conclude that the DCT fragments inhibit neuritogenesis through actions on Cav1.3 activation and inactivation. This is problematic first because the possibility that the DCT fragments regulate Cav1.2-dependent signaling cannot be discounted. The pharmacological dissection of Cav1.3 is not convincing: in Supp Fig.2, the authors show that a “cocktail” of inhibitors including 10 μM nimodipine incompletely blocks the whole cell Ica. Because the L-type blocker isradipine (20 μM) but not the R-type blocker SNX-482 blunts the residual current, the authors conclude this approach effectively isolates the Cav1.3 contribution. The data are derived from drug effects on Ica evoked by only one test pulse. Because dihydropyridine block is voltage-dependent, analysis at a single pulse could underestimate the contribution of Cav1.2 to the whole cell Ica in the cortical neurons in Supp fig2A, such that Cav1.2+Cav1.3 mediate the isradipine-sensitive current. Cav1.3 KO or DHP-insensitive Cav1.2 mutant mice are available and could readily address this caveat (there is also the wrinkle caused by splice variation of Cav1.2 and Cav1.3 affecting dihydropyridine sensitivity in neurons). Fig. 2A shows the effects of the DCRDF on Ica evoked by -10 mV, and the authors conclude that the reduction in the current amp is consistent with the effects of*

DCRDF on inhibiting activation. The authors cannot discount that DCRDF affects channel trafficking to the membrane-- why not show the IV o back up their interpretation? This is a concern since G. Pitt's group showed that blunting CaM interactions with Cav1 can inhibit cell-surface and dendritic trafficking these channels in neurons.

First of all, the goal of our study is to link the DCT inhibition on Cav1 (here focusing on Cav1.3) with the (cytosolic) effects of DCT on neuritogenesis, but with no intention to exclude any potential contribution from Cav1.2. Within the voltage range of low membrane depolarizations (~-20 mV and below), Cav1.3 channels are considered to make substantial or even predominant contributions (as compared to Cav1.2 and other Cav channels) to the Ca²⁺ influx/CaMKII/CREB signaling axis, documented in the literature (Wheeler 2012 Cell, PMID: 22632974; Zhang 2006 Eur J Neuroscience, PMID: 16706838; Zhang 2005 J Neuroscience, PMID: 15689539). For validation purposes at the major checkpoint, we conducted our own experiments to estimate the relative contribution of Cav1.3 versus Cav1.2 in cortical neurons (**Supplementary Fig. 2**). Collectively, we fully agree with the reviewer that Cav1.2 should resemble Cav1.3 in the aspects of DCT/CMI modulation in neurons, but likely to a lesser extent, and the precise difference may depend on various factors such as development stages and neuron subtypes.

In **Fig. 2** and **Fig. 4**, DCT peptides from Cav1.1-1.4 were systematically examined for a common scheme (tested with Cav1.3 channels), which is expected to expand onto the whole channel family of Cav1 since they share similar "calmodulation" (Ben-Johny 2014 J Gen Phy, PMID: 24863929). Also, to address this concern raised by the reviewer, we examined the effects of DCT_F on Cav1.2 in particular, which has proved such applicability (**Supplementary Fig. 16**). Note that both inactivation and activation were inhibited, while to a lesser extent than Cav1.3 presumably due to weaker PCRDC unveiled in this study (**Fig. 4**). For the same reason, CMI by neuronal DCT_C or DCT_D peptides is expected to be less potent for Cav1.2 channel in neurons, as another factor to focus on Cav1.3 for DCT effects.

Next, we would like to address the comments more specifically.

1. Cav1.2 and Cav1.3 in cortical neurons
 - a) For Cav1-mediated signaling (pCREB etc.) in neurons, Cav1 is advantaged over Cav2 in large part due to its lower activation threshold (Wheeler 2012 Cell, PMID: 22632974). Similar principles of preferred channels should be applicable to Cav1.2 and Cav1.3 which have about 10 mV or more difference in activation. The voltage dependence of Cav1.3 is more negative thus preferred for the downstream signaling among other factors (Zhang 2006 Eur J Neuro, PMID: 16706838; Zhang 2005 J Neuroscience, PMID: 15689539). Supported by multiple lines of evidence including our electrophysiology, western-blot and immunostaining data (**Fig. 1, Fig. 6, Supplementary Fig. 2, Supplementary Fig. 12 and Supplementary Fig. 13**), Cav1.3 contributes to a substantial amount/percentage of the total Cav1 channels.

Therefore, Cav1.3 would play a significant or even predominant role in Cav1-mediated signaling and neuritogenesis, especially for neurons under basal conditions. In this work, CMI/DCT effects were unveiled by focusing on Cav1.3, which in principle should be expandable onto Cav1.2 channels, supported by our data (**Supplementary Fig. 16**), and also the previous reports (Sang 2016 Nature Communications, PMID: 27456671; Kamijo 2018 JNS, PMID: 29773754). The relative contributions of Cav1.2 versus Cav1.3 await future investigations, in the context of DCT peptide effects in cortical neurons.

- b) Detailed analyses on Ca²⁺-dependent inactivation validate our approach to isolate Cav1.3 currents in cortical neurons (**Fig. 1a**). Data and analyses on Cav1.2, DCT_c and PCRD_c suggest that Cav1.2 channels in neurons normally produce rather strong inactivation due to its weak PCRD_c. However, in our recordings with the full cocktail recipe, the residue Ca²⁺ current (*I_{Ca}*) exhibited moderate to weak inactivation, indicative of its dominance of Cav1.3 considering Ca²⁺-dependent inactivation is a rather robust index in practice. In support, from cortical neurons overexpressed with CaM (unpublished data), the above *I_{Ca}* (exactly the same cocktail recipe) resulted in much stronger inactivation (*S_{Ca}* increased from 0.4 to 0.7, n=6), a stereotypical feature of Cav1.3. In contrast, with recombinant Cav1.2 and overexpressed CaM (HEK cells), only slight changes can be further induced on top of inactivation that is already very strong (*S_{Ca}* from 0.57 to 0.65, n=7). Altogether, although we agree with the reviewer that additional methods and data should provide more precise and thorough analyses, the information thus far has sufficiently justified Cav1.3 as the representative channels to focus on in this study.
- c) Additional clarifications. In the revised **Supplementary Fig. 2**, only isradipine data of 20 μM is included for clarity in light of the reviewer's comments. Also, the pharmacological recipe is one viable option here. Notably, precautions are needed for using KO mice since Cav1 channels play essential roles in gene transcription and neuronal development, and also encode the DCT peptides. KO mice lacking critical genes such as Cav1 may or may not provide the proper and reliable information for the targeted proteins and signals. More delicate approaches, such as the knock-in mice with DHP-insensitive Cav1.2 and/or Cav1.3 mutations as the reviewer suggested, should be employed for a rigorous comparison, e.g., to compare channel currents under acute DHP administration.

2. Effects on surface Cav1.3

Please see our response to Reviewer #1. The previous studies (Liu 2017 eLife, PMID: 28059704; Adams 2014 Cell, PMID: 25417111) and this work (**Supplementary Fig. 14c**) support that for Cav1.3 channels, DCT/CaM competition is an acute modality of channel inhibition, with no or negligible contribution from any potential effect on surface expression or membrane trafficking of Cav1.3 channels.

Note that additional effects through alternative mechanisms (other than acute CMI) are still possible for Cav1.2 as mentioned by the reviewer (Wang 2007 JNS, PMID: 17715345). In fact, the previous reports on surface expression or membrane trafficking in relation to CaM and IQ are mostly based on Cav1.2, inviting the question whether DCT peptides could reduce the (surface) expression of Cav1.2. It would be interesting to explore the potential difference between Cav1.3 versus Cav1.2 in this aspect.

2) Despite the focus on Cav1.3, the neuronal experiments utilize strategies that would not distinguish between Cav1.2 and Cav1.3. In Figs 2C,D, high K⁺ (which will activate both Cav1.2 and Cav1.3) is used to analyze nuclear CaM and pCREB. How this signaling pathway might contribute to neuritogenesis analyzed in Fig. 2B and c-fos (fig. 2E) is not clear (these assays did not utilize high K).

Please see item 1 (“**Basal** versus **excited**”) in the introductory part of our response. In this study, we have been focusing on Cav1.3, the exemplary Cav1 channels in cortical neurons, considering that Cav1.3 is advantaged over Cav1.2 particularly due to the “gating advantage” under the basal conditions. For clarity, we now have reorganized the relevant signaling data, so that all the main figures are under basal conditions consistently (**Fig. 1** and **Fig. 6**), while high K⁺ data now go to the supplemental (**Supplementary Fig. 3**). Note that **Fig. 1c** is newly provided for the effects of DCT on basal-level pCREB.

In fact, regardless of the states (basal or excited), the following signaling axis should be well functioning in neurons: Cav1 activation, Ca²⁺ influx, pCaMKII, CaM/CaMKII translocation, pCREB, and essential genes such as c-Fos, which has been manifested by Cav1.3 and cortical neurons under basal conditions in this work. Cav1.2 may also play a significant role in certain scenarios of cell signaling (e.g., in strongly excited neurons), where it is possible for Cav1.2 to make substantial contributions to CMI effects on neuritogenesis. We have no intention to exclude Cav1.2, just that Cav1.3 is more advantaged over Cav1.2 in the context of our work. The precise contributions of Cav1.2 versus Cav1.3 need to be quantified in the future.

3) There is rather loose citing of past work in support of the authors' interpretations. Redmond, Ghosh show that Cav1-mediated excitation-transcription coupling regulates activity-dependent neurite growth (i.e., that promoted by high K⁺). However, the paradigm here looks at effects of DCT fragments on basal neurite growth. Krey, Dolmetsch show that a Timothy syndrome mutation that suppresses inactivation causes dendrite retraction as opposed to the stimulatory effects of DCT peptides on neurite growth studied here.

Cav1 channels under basal conditions are already activated to certain levels (e.g., manifested as Cav1-dependent autonomous oscillations), which make significant contributions to “basal”

neuritogenesis (Kamijo 2018 JNS, PMID: 29773754). Please note that we refer to “activities” here as channel activities, which include but are not limited to the activities associated with extrinsic/extracellular excitation (such as sensory stimuli or high K⁺). In the context of **calcium channel activity-dependent neuritogenesis**, basal or elevated levels of channel activities have no fundamental difference, which share the common signaling cascade. Considering the critics and also the essence of our study, we have revised the title of our manuscript as “Cytosolic peptides encoding Cav1 C-termini downregulate the **calcium channel activity-neuritogenesis coupling**”.

Cav1 modulation or modulators traditionally target the Ca²⁺ influx, with the assumption that Cav1 signaling (of the excitation-transcription coupling or the channel activation-neuritogenesis coupling) would be modulated accordingly. Although in many cases Ca²⁺ influx and downstream signals are indeed consistently regulated, different lines of evidence suggest that the two (influx versus signaling) may often decouple from each other, known as flux-independence. For example, voltage (induced protein conformation) is also crucial to drive pCREB in addition to Ca²⁺ influx (Li 2016 Science, PMID: 26912895). Similarly, instead of neurite extension normally expected from an increase of Cav1 influx, TS gain-of-function mutations lead to neurite retraction due to ectopic GEM/RhoA signaling (Krey 2013 Nature Neuroscience, PMID: 23313911). Similar discrepancies of influx-signals decoupling have been reported for chemical agonists or antagonists.

To clarify the fact, DCT peptides do have the “*stimulatory*” effects on neurite outgrowth when present in the nucleus (Gomez-Ospina 2006 Cell, PMID: 17081980); and our discovery in this work is the opposing role of DCT peptides in the cytosol: **inhibition** of neuritogenesis.

4) Relevant to the previous point, pharmacological and genetic suppression of Cav1 activity is known to promote rather than inhibit neurite regeneration in some contexts (PMID: 23313911, 20579880, 30232777, 28320840). The authors should reconcile their findings with this past work.

In addition (to the last response), a variety of other factors, e.g., neuron subtypes, development stages, disease states, etc., could further complicate the situations. Thus, no simple conclusion could be reached to directly reconcile the discrepant findings in the literature. In part due to this reason, DCT peptides are very unique in our opinion, which consistently inhibit the whole cascade, starting from Ca²⁺ influx, via multiple intermediate signals, all the way to neurite outgrowth, in hope to provide a new toolbox with both potency and specificity.

Based on the literature and our own data, several scenarios are listed as follows:

a) Intact Cav1 channels (e.g., CaM/IQ binding is intact) mediate the Cav1 activity-dependent neuritogenesis through the classical signaling cascade of excitation-

transcription coupling, where Cav1 channels appear to be **constructive** (Dolmetsch 2003 Sci STKE, PMID: 12538881; Deisseroth 2004 Neuron, PMID: 15157417);

b) Cav1 channels are altered or perturbed in a way leading to inhibition of Ca²⁺ signaling and neuritogenesis. Certain antagonists (such as nimodipine) may be able to attenuate Cav1-dependent “constructive” signals underlying neuritogenesis. DCT peptides appear to work this way: Cav1 channels are switched from CaM-bound to DCT-bound states, shutting down pCREB and so on;

c) Furthermore, Cav1 channels, under certain circumstances (e.g., genetic or chemical perturbations), may produce additional **destructive** effects on neuritogenesis, e.g., triggering pruning or retraction signals in concert with particular signaling proteins (Kanamori 2013 Science, PMID: 23722427; Krey 2013 Nature Neuroscience, PMID: 23313911). This scenario awaits future investigations to examine whether Cav1 of destructive mode may contribute to the observed inhibitory effects of DCT/CMI peptides.

These modes of action altogether should help reconcile the complicated or even apparently contradictory findings, such as the four research articles mentioned by the reviewer (**PMID: 23313911, 20579880, 30232777, 28320840**):

a) Triggering destructive Cav1 signals to impair neuritogenesis.

More/less channel activities (thus more/less Ca²⁺ influx) lead to facilitation/inhibition of neurite outgrowth, if the normal (constructive) mode of Cav1 can be assumed. However, certain genetic defects of Cav1, e.g., TS mutations that appear to increase Ca²⁺ influx, may severely distort the conventional Cav1 signaling, or even ectopically trigger destructive signals, thus jeopardize neuronal development (**PMID: 23313911**).

b) Inhibiting destructive Cav1 to promote neuritogenesis.

Suppression of Cav1 channel activities helps axon regeneration/regrowth in injured adult DRG neurons (**PMID: 20579880, 30232777**). One may speculate that harmful events or signals are particularly associated with the destructive activities of altered channels, therefore if such activities or signals are suppressed, it would alleviate or even revert the damages. Reportedly, knock-down of Kv3.4 impedes axon development by promoting Ca²⁺ influx via Cav1 (**PMID: 28320840**), potentially of destructive modes as an attractive hypothesis to follow up in the future.

We agree with the reviewer that these questions are crucial, and we have revised the text accordingly (Page 18-Line 499).

5) Results in Fig.6 investigating the nuclear localization dynamics of the endogenous CTD peptide are compromised by the lack of rigor in characterizing the specificity of the CTD antibodies. Western blot in Fig.6A showing band detected in mouse brain but not untransfected CHO cells does not discount possibility that what is detected by IF in B-D is unrelated to Cav1.3 (the paper cited in the methods for characterizing the Cav1.3 antibody used did not use appropriate negative controls for specificity). The authors imply that the 55-kDa fragment results from proteolysis of the channel, but this has not been shown definitively

for mouse cortical neurons.

Regarding the technical concern on the specificity of CT_D antibody, we have provided additional control samples: liver, 3xFlag-DCT_D and 3xFlag-DCT_C (new **Supplementary Fig. 12**). No crosstalk has been found in our experiments. The control antibody of anti-CT_C was compared with anti-CT_D in this revision (new **Fig. 6a-c**). Briefly, in neurons, Cav1.3 DCT peptides were confirmed to be present, shown by the western-blot band of ~60 kDa (for simplicity, the long-form peptides are now referred to as ~60 kDa); in parallel, DCT_C peptides were also detected by anti-CT_C, resulting in two bands: ~60 kDa and ~40 kDa. Moreover, such DCT peptides (potentially together with their production mechanisms) appeared to be well conserved in HEK293 cells. Similar results were obtained: Cav1.3-encoded peptide of ~60 kDa and Cav1.2-encoded peptides of ~60 kDa and ~40 kDa, after Cav1.2 and Cav1.3 of full-length (both tagged with Flag) were transfected into HEK cells, detected by both Flag and anti-CT_{C/D}.

To further consolidate the presence of endogenous CT_D in cortical neurons, additional immunostaining results have been supplied with Cav1.3 antibody targeting its extracellular loop besides CT_D antibody (new **Supplementary Fig. 13**). In cortical neurons, nuclear fluorescence of high intensity is often detected by CT_D antibody, but not by the antibody of extracellular loop, supporting the presence of nuclear CT_D fragments.

Cav1.3-encoded DCT fragments in cells were firstly reported by Lu 2015 JBC, PMID: 25538241. Here we provide the first piece of evidence that Cav1.3 DCT fragments also exist in cortical neurons of neonatal mice. Besides neurons, such CT_D peptides could also be produced in the recombinant system (overexpressing full-length α_{1D} in cell lines). However, it is unclear to us where the reviewer got the impression "*The authors imply that the 55-kDa fragment results from proteolysis of the channel*" (although functionally CT_D and CCT_D are supposed to resemble each other). To our understanding, it needs careful and systematic work to identify the mechanisms of production for any Cav1-encoded peptides. Due to this uncertainty, we used a different name of CT_D other than any existing peptides including CCT_D. For CCT_D, even named as cleaved peptides, it is still up to future evidence to prove or disprove whether the peptides (~60 kDa band) are actually generated by proteolysis in cardiac myocytes (Lu 2015 JBC, PMID: 25538241). Here, we would like to provide our speculations. New data were acquired for both Cav1.3 and Cav1.2 in both neurons and HEK cells for this revision. For CT_C, consistent with the published results (Du 2019 Neuron PMID 30922876), we observed a similar size of ~60 kDa band from both recombinant systems and cortical neurons, consistent with a conserved mechanism of production, i.e., bicistronic transcription (other than proteolysis). Likewise, we found the CT_D band of ~60 kDa encoded by endogenous or exogenous Cav1.3, potentially through the same bicistronic mechanism as Cav1.2 (new **Fig. 6** and **Supplementary Fig. 12**). Again, dedicated future work is in need to conclude on bicistronic production of Cav1.3 DCT peptides.

Finally, for the cited work using anti-CT_D (Chovancova 2019 J BCP, PMID: 31706846) and also the reference of this antibody on the website abcom.com (Gao 2020 J Int Med Res, PMID: 32812463), it is true that no control has been provided thus lack of rigor. In response to this concern, we have conducted additional (control) experiments to resolve the confusions and doubts on Cav1.3-encoded peptides in cortical neurons. We are very grateful to the reviewer for the insightful comments and suggestions.

6) *Statistical analysis is improperly documented and sometimes absent (i.e., supp Fig.2, all Sholl analyses). Tests to verify normality of the data are warranted for the tests used (t-tests, ANOVA).*

By applying D'Agostino & Pearson omnibus normality test, we have verified that all the data comply with the normal distribution except those related to c-Fos. For these particular data sets (**Fig. 1d** and new **Supplementary Fig. 14b**), the statistical analysis has been replaced with the Kruskal-Wallis and Dunn's non-parametric test. In **Supplementary Fig. 2**, data in the form of Mean \pm -SEM should provide sufficient information for the purpose of an approximate estimation on the relative contribution of Cav1.3 in the particular setting of experiments, since all these channel blockers are well-established in the literature. Please note that the numbers of cells have been routinely supplied to help address the concern of statistics.

For Sholl analyses, we have appended the index of AUC (area under the curve) for more rigorous quantification/statistics as the reviewer suggested (**Fig. 1b**, **Fig. 3c**, **Fig. 5c**, **Fig. 5g**, **Supplementary Fig. 9b** and **Supplementary Fig. 11f**).

Minor comments:

1) *Include the number of independent cultures per experiment along with the number of cells from each cortical neuron culture. The cell numbers are different in all experiments, so it is hard to follow the cell numbers from each independent culture.*

We have stated explicitly about the number of independent cultures, experiments, and cells in the Methods, separately for morphology imaging (Line 758), immunocytochemistry (Line 791) and western blot (Line 816) experiments. Please note that for certain data less cells than the above guidelines may be indicated (e.g, **Fig. 3**), which is due to that the total cells were further divided into subgroups (cytosolic versus nuclear). For these data, the analyses and conclusions are subsequently confirmed by fusing NES- and NLS-tags.

2) *Sholl analysis is often compared as the area under the curve (AUC) which would help back up claims re: significant differences in Fig. 1b, Fig. 3c and Fig. 5c, g*

We have revised accordingly, please see our response to Major comment 6.

3) *Text could use editing for grammar.*

We have revised the text according to this suggestion.

4) *Double check text matches the figure references. Page 13, line 362 has a mislabeled figure number it reads (Fig. 6i, j) which I think refers to (Fig. 5i, j). Same mistake in page 15, line 414. Page 20, line 556 refers to supplementary Fig. 9 for the distinctions between the short and long/medium peptides demonstrate but that figure only shows the N/C ratio of DCRDC/D. I think Fig 5 h and I are the proper figure reference.*

We have revised the text according to the comments. And for the last item regarding the distinction between the short versus long/medium peptides, it has been revised as (**Fig. 5h, i** and **Supplementary Fig. 10**, Page 21-Line 597). The presence/absence of NRD domain is the key to the observed differences between the short DCRD_{C/D} peptides and other longer peptides.

Reviewers' comments:

Reviewer #1 (Remarks to the Author):

The authors are to be commended for their rigorous approach to addressing concerns raised in the critiques of the original manuscript. The authors answered all my concerns.

Reviewer #3 (Remarks to the Author):

While the authors have addressed some key concerns, their conclusions regarding the involvement of Cav1.3 is still weakly supported. In their rebuttal, the authors cite Supp Fig.2, Supp Fig.12, Supp fig.13 as evidence that "CaV1.3 contributes to a substantial amount/percentage of the total CaV1 channels".

Regarding Supp Fig.2: "Evaluated at the step of -10 mV representing the CaV1 dominant voltage range (Wheeler et al., 2012) CaV1.3 made a significant contribution to the total CaV 1 currents in cortical neurons, consistent with the earlier reports proposing a predominant role of Ca V 1.3 (versus CaV1.2) in CaV1 signaling at relatively low levels of depolarization (Zhang et al., 2005, 2006).

First, the relevant experiments of the Wheeler et al. study were done in superior cervical ganglion neurons whereas the Zhang et al studies utilized heterologous expression of Cav1.2 and Cav1.3 constructs in *Xenopus* oocytes and hippocampal neurons. Given that the voltage-dependence of Cav1 activation will vary with external divalent concentration (PMID 16267232) as well as alternative splicing of the Cav1.2 and Cav1.3 mRNAs (PMID 18482979, 15299022, 21998309), it is dicey to extrapolate the behavior of Cav currents evoked by single test pulses based on voltage-dependence established in other systems. Supp Fig.2 would be far more convincing if the residual current blocked by isradipine (i.e., Cav1.3) shows an IV with left-shifted voltage-dependence of activation. In lieu of that, a less confusing presentation in Supp Fig.2 would help. To back up the data summarized in the pie charts in panel D, current traces should be shown to illustrate block of the different Cav subtypes in neurons: +MVIIC+GVIA (Cav2.1, 2.2), +MVIIC+GVIA+nimodipine (Cav2.1, Cav2.2, Cav1.2), +MVIIC+GVIA+nimodipine+isradipine (Cav2.1, Cav2.2, Cav1.2, Cav1.3). The bar graph in panel C should contain only the data from the neurons so that statistical comparisons could be made illustrating the significance of the blockers in terms of blunting the various Cav components (including the HEK cell data in the current figure was very confusing, especially since the legend does not clarify the origin of those data).

Regarding Supp Fig.12,13: The western blots show that CT antibodies work well for overexpressed channel proteins, but do not address specificity in the context of the experiments in the main paper (i.e., neurons). This is critical given the notorious non-specific nature of Cav antibodies and that these proteins are expressed at low levels compared to other proteins that may be nonspecifically recognized. Documentation of specificity is entirely lacking for the extracellular loop antibody used in Supp. Fig.13, 14c. In my view, removal of the data involving these antibodies would not be detrimental to the overall conclusions of the paper, and would help focus attention on the very nice structure/function analyses. In lieu of convincing lack of immunoreactivity in Cav1.3 KO neurons, the authors should acknowledge the limitations of these data.

Regarding the point raised by me and reviewer 1 that CCT peptides might affect membrane protein expression of the channel, I'd backtrack on that concern given the difficulty of confirming antibody specificity if the authors could show more convincingly that what is shown in Fig.2b (full IV if not the r50 v voltage curves) is recapitulated in the neurons.

[Reviewers' comments in *Italic* fonts and our responses in normal fonts]

Reviewer #1 (Remarks to the Author):

The authors are to be commended for their rigorous approach to addressing concerns raised in the critiques of the original manuscript. The authors answered all my concerns.

We would like to express our gratitude to the reviewer, for the encouragement and the professional service through the whole reviewing process.

Reviewer #3 (Remarks to the Author):

1) While the authors have addressed some key concerns, their conclusions regarding the involvement of Cav1.3 is still weakly supported. In their rebuttal, the authors cite Supp Fig.2, Supp Fig.12, Supp fig.13 as evidence that “CaV1.3 contributes to a substantial amount/percentage of the total CaV1 channels”.

Regarding Supp Fig.2: “Evaluated at the step of -10 mV representing the CaV1 dominant voltage range (Wheeler et al., 2012) CaV1.3 made a significant contribution to the total CaV 1 currents in cortical neurons, consistent with the earlier reports proposing a predominant role of Ca V 1.3 (versus CaV1.2) in CaV1 signaling at relatively low levels of depolarization (Zhang et al., 2005, 2006).

First, the relevant experiments of the Wheeler et al. study were done in superior cervical ganglion neurons whereas the Zhang et al studies utilized heterologous expression of Cav1.2 and Cav1.3 constructs in Xenopus oocytes and hippocampal neurons. Given that the voltage-dependence of Cav1 activation will vary with external divalent concentration (PMID 16267232) as well as alternative splicing of the Cav1.2 and Cav1.3 mRNAs (PMID 18482979, 15299022, 21998309), it is dicey to extrapolate the behavior of Cav currents evoked by single test pulses based on voltage-dependence established in other systems. Supp Fig.2 would be far more convincing if the residual current blocked by isradipine (i.e., Cav1.3) shows an IV with left-shifted voltage-dependence of activation. In lieu of that, a less confusing presentation in Supp Fig.2 would help. To back up the data summarized in the pie charts in panel D, current traces should be shown to illustrate block of the different Cav subtypes in neurons: +MVIIC+GVIA (Cav2.1, 2.2), +MVIIC+GVIA+nimodipine (Cav2.1, Cav2.2, Cav1.2), +MVIIC+GVIA+nimodipine+isradipine (Cav2.1, Cav2.2, Cav1.2, Cav1.3). The bar graph in panel C should contain only the data from the neurons so that statistical comparisons could be made illustrating the significance of the blockers in terms of blunting the various Cav components (including the HEK cell data in the current figure was very confusing, especially since the legend does not clarify the origin of those data).

In light of these suggestions, we have conducted patch-clamp recordings to address the question regarding the actual components underlying Cav1 currents in cortical neurons. In summary, we have confirmed that: 1) Cav1.3 makes a significant contribution to the total Cav1 currents: about equal or slightly more in comparison with Cav1.2 according to our measurements, analyses and estimations; 2) the voltage-dependent activation of residue (Cav1.3) current is left-shifted in comparison with that of Cav1.2 or Cav1.2-dominant currents. In this revision, the amplitude and the half-activation voltage (V_{half}) were evaluated from the full voltage range from -60 mV to +50 mV. More details can be found in the updated **Supplementary Fig. 2**.

Also, we would like to thank the reviewer for the insights toward the literature regarding Cav1.2 versus Cav1.3, and we agree with nearly all of these points. In cortical and other neurons Cav1.2 has been more extensively characterized or studied. However, this might in part reflect the fact that historically Cav1.2 caught earlier and more attention than Cav1.3. In fact, Cav1.3 is equally abundant in cortex unveiled by RT-PCR, immunostaining or WB (PMID: 26459417; PMID: 8227151 and PMID: 17823125).

2) Regarding Supp Fig.12,13: The western blots show that CT antibodies work well for overexpressed channel proteins, but do not address specificity in the context of the experiments in the main paper (i.e., neurons). This is critical given the notorious non-specific nature of Cav antibodies and that these proteins are expressed at low levels compared to other proteins that may be nonspecifically recognized. Documentation of specificity is entirely lacking for the extracellular loop antibody used in Supp. Fig.13, 14c. In my view, removal of the data involving these antibodies would not be detrimental to the overall conclusions of the paper, and would help focus attention on the very nice structure/function analyses. In lieu of convincing lack of immunoreactivity in Cav1.3 KO neurons, the authors should acknowledge the limitations of these data.

In response to the comments and suggestions, we have reorganized the logic of the main text, from which the WB data using Cav1-related antibodies are detached. Data using Cav1 CT antibodies are now moved into the supplementary figures, solely serving as the supportive evidence for DCT effects demonstrated by exogenous peptides in cortical neurons. Also, the immunostaining data by Cav1.3 extracellular-loop antibody (alomone labs, #ACC311) previously shown in **Supplementary Fig. 13** and **14c** have been completely removed from the manuscript.

Meanwhile, we would like to share some of our thoughts on the limitations of our work.

- a) The antibody ACC-311 (epitope: the extracellular loop of Cav1.3) is indeed in lack of sufficient support, compared with other more popular Cav1 antibodies. As

the reviewer pointed out, “this is critical given the notorious non-specific nature of Cav antibodies and that these proteins are expressed at low levels compared to other proteins that may be nonspecifically recognized”. In the future, rigorous validations, e.g., with Cav1.3 knock-out (ideally conditional knock-out) neurons and/or the control of blocking peptides, are expected to confirm the WB and immunostaining data using ACC-311 of ours and others (PMID: 29079724; PMID: 28585545; PMID: 27445440).

- b) Multiple methods instead of relying on one single approach should be an effective strategy, especially by focusing on some unique checkpoint(s) other than diverting the efforts to excessive targets/checkpoints/events. To address the particular question regarding the functional expression of Cav1.3 (and the DCT effects), a set of consistent data obtained from electrophysiology, biochemistry and imaging with multiple peptide variants should lead to more convincing conclusions. Single approach (such as Cav1 antibodies) is often found to be supportive, instead of providing more conclusive insights. Our work is also being limited in this aspect.
- c) We agree with the reviewer that Cav1.3 antibodies should be more rigorously examined, by taking advantage of transgenic mice as suggested. However, due to the compensatory effects on Ca²⁺ channels in Cav1.3^{-/-} and Cav1.2^{-/-} mice (PMID: 12900400; PMID: 11581302), cautions need to be taken in using these knock mice and interpreting the data from these mice. Some alternative approach(es) may be necessary. For example, regarding the isolation of Cav1.3 versus Cav1.2 currents, it would be better to employ knock-in mice including Cav1.2 DHP^{-/-} and/or Cav1.3 DHP^{-/-}. With such preparations, we would gain the access to the “true” Cav1.3 and Cav1.2 currents in full, followed by thorough characterizations (full-range voltage steps with both Ba²⁺ and Ca²⁺, standard analysis of both activation and inactivation, single channel recordings, etc.).

We have incorporated these limitations of our work into the revised Discussion (Page 21, Line 571-581).

3) Regarding the point raised by me and reviewer 1 that CCT peptides might affect membrane protein expression of the channel, I'd backtrack on that concern given the difficulty of confirming antibody specificity if the authors could show more convincingly that what is shown in Fig.2b (full IV if not the r50 v voltage curves) is recapitulated in the neurons.

Regarding the full I-V curves for cortical Cav1.3 currents, please see the updated **Supplementary Fig. 2**. Even without the supportive immunostaining data previously obtained with Cav1.3 antibodies, there are at least two additional lines of evidence.

- a) For recombinant Cav1.3 channels, the peptide effects (CMI) have been confirmed to be acutely inducible, which is nearly identical to the effects by the peptides constitutively expressed in the cells (for two days), suggesting that there is unlikely any chronic effect (such as reduction of trafficking to plasma membrane) of DCT peptides (Liu 2017 eLife, PMID: 28059704).
- b) Mechanistically, DCT and apoCaM have opposing effects on Cav1.3. No change in Cav1.3 expression has been evidenced when overexpressing CaM in the recombinant system (Adams 2014 Cell, PMID: 25417111), or CaM knock-down and rescue in cortical neurons (Pang 2010 JBC, PMID: 20729199). All these facts are consistent with the notion that the competition between DCT and CaM would not alter the surface expression of functional Cav1.3 channels.

REVIEWERS' COMMENTS:

Reviewer #3 (Remarks to the Author):

I appreciate the authors' attention to my suggestions and feel that they have sufficiently addressed my concerns. This manuscript should provide important new information regarding the structure/function relations of Cav1 channels and their physiological roles.